# Epidermal LysM receptor ensures robust symbiotic signalling in *Lotus japonicus*

Eiichi Murakami[1], Jeryl Cheng[1], Kira Gysel[1], Zoltan Bozsoki[1],
Yasuyuki Kawaharada[1†], Christian Toftegaard Hjuler[2],
Kasper Kildegaard Sørensen[2], Ke Tao[1], Simon Kelly[1], Francesco Venice[3],
Andrea Genre[3], Mikkel Boas Thygesen[2], Noor de Jong[1], Maria Vinther[1],
Dorthe Bødker Jensen[1], Knud Jørgen Jensen[2], Michael Blaise[1‡],
Lene Heegaard Madsen[1], Kasper Røjkjær Andersen[1], Jens Stougaard[1],
Simona Radutoiu[1]*

[1]Department of Molecular Biology and Genetics, Aarhus University, Aarhus, Denmark; [2]Department of Chemistry, University of Copenhagen, Frederiksberg, Denmark; [3]Department of Life Sciences and Systems Biology, University of Torino, Torino, Italy

*For correspondence: radutoiu@mbg.au.dk

Present address: †Department of Plant Bio Sciences, Faculty of Agriculture, Iwate University, Morioka, Japan; ‡IRIM-UMR 9004, Research Institute in Infectiology of Montpellier, University of Montpellier, CNRS, Montpellier, France

**Abstract** Recognition of Nod factors by LysM receptors is crucial for nitrogen-fixing symbiosis in most legumes. The large families of LysM receptors in legumes suggest concerted functions, yet only NFR1 and NFR5 and their closest homologs are known to be required. Here we show that an epidermal LysM receptor (NFRe), ensures robust signalling in *L. japonicus*. Mutants of *Nfre* react to Nod factors with increased calcium spiking interval, reduced transcriptional response and fewer nodules in the presence of rhizobia. NFRe has an active kinase capable of phosphorylating NFR5, which in turn, controls NFRe downstream signalling. Our findings provide evidence for a more complex Nod factor signalling mechanism than previously anticipated. The spatio-temporal interplay between *Nfre* and *Nfr1*, and their divergent signalling through distinct kinases suggests the presence of an NFRe-mediated idling state keeping the epidermal cells of the expanding root system attuned to rhizobia.
DOI: https://doi.org/10.7554/eLife.33506.001

## Introduction

Perception of Nod factors by LysM receptor kinases, NFR1 and NFR5 in *Lotus japonicus* (*Broghammer et al., 2012*), triggers tightly coordinated events leading to root nodule symbiosis (*Madsen et al., 2003*; *Radutoiu et al., 2003*). Minutes after the activation of receptors, a signalling cascade (*Stracke et al., 2002*; *Antolín-Llovera et al., 2014*) leading to regular calcium oscillations in the root hair cells located in the susceptible zone is initiated (*Miwa et al., 2006*). These oscillations are interpreted by the Calcium Calmodulin Kinase (CCaMK) (*Miller et al., 2013*), which activates a set of regulators that launch transcription of symbiosis specific genes in the outer root cell layers (*Yano et al., 2008*; *Hirsch et al., 2009*). Progression of the symbiotic signalling events from epidermis into the cortex is necessary for nodule organogenesis and infection thread formation. NIN, a transcriptional regulator, and cytokinin signalling have been implicated in this epidermal to cortex signalling (*Murray et al., 2007*; *Tirichine et al., 2007*; *Vernié et al., 2015*).

Mutations in *Nfr5* and its homologs in pea and *M. truncatula* eliminate all Nod factor-induced physiological, molecular and cellular responses (*Madsen et al., 2003*; *Arrighi et al., 2006*). However, some or several of these responses are retained in the *Ljnfr1*, *Mtlyk3* and *Pssym37* mutants (*Radutoiu et al., 2003*; *Smit et al., 2007*; *Zhukov et al., 2008*) raising the possibility that modular receptor complex formation regulated in a spatio-temporal manner might contribute to Nod factor

**eLife digest** Microbes – whether beneficial or harmful – play an important role in all organisms, including plants. The ability to monitor the surrounding microbes is therefore crucial for the survival of a species. For example, the roots of a soil-growing plant are surrounded by a microbial-rich environment and have therefore evolved sophisticated surveillance mechanisms.

Unlike most other plants, legumes, such as beans, peas or lentils, are capable of growing in nitrogen-poor soils with the help of microbes. In a mutually beneficial process called root nodule symbiosis, legumes form a new organ called the nodule, where specific soil bacteria called rhizobia are hosted. Inside the nodule, rhizobia convert atmospheric dinitrogen into ammonium and provide it to the plant, which in turn supplies the bacteria with carbon resources.

The interaction between the legume plants and rhizobia is very selective. Previous research has shown that plants are able to identify specific signaling molecules produced by these bacteria. One signal in particular, called the Nod factor, is crucial for establishing the relationship between these two organisms. To do so, the legumes use specific receptor proteins that can recognize the Nod factor molecules produced by bacteria. Two well-known Nod factor receptors, NFR1 and NFR5, belong to a large family of proteins, which suggests that other similar receptors may be involved in Nod factor signaling as well.

Now, Murakami et al. identified the role of another receptor called NRFe by studying the legume species *Lotus japonicus*. The results showed that NFRe and NFR1 share distinct biochemical and molecular properties. NRFe is primarily active in the cells located in a specific area on the surface of the roots. Unlike NFR1, however, NFRe has a restricted signaling capacity limited to the outer root cell layer. Murakami et al. found that when NRFe was mutated, the Nod factor signaling inside the root was less activated and fewer nodules formed, suggesting NRFe plays an important role in this symbiosis.

NFR1-type receptors have also been found in plants outside legumes, which do not form a symbiotic relationship with rhizobia. Identifying more receptors important for Nod-factor signaling could provide a basis for new biotechnological targets in non-symbiotic crops, to improve their growth in nutrient-poor conditions.

DOI: https://doi.org/10.7554/eLife.33506.002

signalling. The LysM receptor kinase family has greatly expanded in legumes through whole genome or tandem duplications (*Zhang et al., 2009*; *Lohmann et al., 2010*; *Kelly et al., 2017*). In *L. japonicus*, four genes, *Lys1*, *Lys2*, *Lys6* and *Lys7*, are closely related to *Nfr1*. *Lys1* and *Lys2* are located in tandem and at ~10 kb distance from *Nfr1* (*Lohmann et al., 2010*). Interestingly, a similar chromosomal organisation of NFR1-type receptors was reported in all studied legumes, as well as in genomes outside of *Leguminosae* clade raising the possibility that gene duplication leading to tandem NFR1-type receptors preceded the evolution of the legume family (*De Mita et al., 2014*). The precise role of these NFR1 paralogs and their homologs is unknown apart from the chitin receptors *L. japonicus* LYS6 (now CERK6) and *Medicago truncatula* LYK9 (*Bozsoki et al., 2017*). Nonetheless, key details about the signalling competencies of LYS2, LYS6 and LYS7 were obtained from functional complementation analyses in the *nfr1* mutant using the *LjUbiquitin* promoter. Only the intracellular kinase regions of LYS6 and LYS7, but not that of LYS2, could restore nodulation and/or infection when coupled to the NFR1 Nod factor-binding domain (*Nakagawa et al., 2011*).

Here, we show that LYS1 is an epidermal LysM receptor contributing to the NFR1-NFR5 mediated signalling in a spatio-temporal manner. This gene is primarily expressed in epidermal cells of the susceptible zone where roots are competent for initiation of symbiosis, and has a restricted signalling capacity leading to *Nin* activation in the outer root cell layers. Our findings provide evidence for a complex Nod factor signalling where LYS1 activity in the outer root cell layers aids in maintaining a normal calcium spiking interval in the root hairs, integral transcript responses in the susceptible root zone, and initiation of nodule primordia on the expanding root system. The *Lys1* gene is therefore renamed *Nfre*, in accordance with the identified role of this gene during *Nod factor* signalling in the epidermal layer.

## Results

### NFRe perceives Nod factor and has an active intracellular kinase

NFRe is predicted to encode a LysM receptor protein with a typical intracellular kinase domain (*Figure 1A*, *Figure 1—figure supplement 1*). Based on the close similarity to NFR1 we investigated its biochemical and molecular properties. For this, we analysed the binding capacity of NFRe towards purified pentameric *M. loti* R7A Nod factor ligand (*Bek et al., 2010*). The NFRe ectodomain was expressed in insect cells using a recombinant baculovirus induced-expression system (*Kawaharada et al., 2015*). Pure protein was obtained after four steps of purification and the homogeneity was confirmed by size exclusion chromatography (*Figure 1—figure supplement 2*). Biolayer interferometry (BLI) (*Kawaharada et al., 2015*) was used for receptor-ligand affinity measurements since this technique is well suited for handling sparingly soluble hydrophobic compounds like Nod factor. To enable ligand immobilization on streptavidin biosensors *M. loti* R7A Nod factor and chitin pentamer ((GlcNAc)5, CO5) were conjugated to a biotinylated linker using *N*-glycosyl oxyamine chemistry (*Bohorov et al., 2006*; *Villadsen et al., 2017*) (*Figure 1—figure supplement 3*). Affinity measurements showed that the ectodomain of NFRe has the capacity to bind *M. loti* Nod factor with an equilibrium dissociation constant (KD) of 29.1 ± 7.1 μM (*Figure 1B,E*). Next, we tested whether NFRe has the capacity to bind chitin but no signal was observed for CO5 ligands in this system (*Figure 1B*). To test our immobilised ligands, we performed the same binding experiment with purified NFR1 ectodomain (*Figure 1—figure supplement 2*), which gave a KD of 34 ± 6.3 μM to *M. loti* R7A Nod factor and no binding to CO5 (*Figure 1C,E*). As a positive control for our chitin ligand we additionally expressed and purified the *Arabidopsis* CERK1 ectodomain (*Figure 1—figure supplement 2*) and measured an affinity of 59 μM to the immobilized CO5 (*Figure 1D,E*), which is very similar to the previously reported KD of 66 μM measured by isothermal titration calorimetry (*Liu et al., 2012*). In short, our binding studies show that NFRe has the capacity to perceive Nod factor with comparable affinity as seen for the NFR1 and both receptor ectodomains distinguish Nod factor from pentameric chitin ligands in a BLI binding assay (*Figure 1E*). NFRe is a challenging and low expressed protein and further biochemical ligand competition studies are required to fully define the specificity and receptor capacity of NFRe.

Next, we assessed the activity of the intracellular kinase domain of NFRe (*Figure 1—figure supplement 1*). *E. coli*-produced NFRe kinase transphosphorylated the myelin basic protein (MBP) substrate and autophosphorylated (*Figure 1F*, lanes 1–3), showing that NFRe encodes a protein kinase with in vitro activity similar to NFR1 (*Madsen et al., 2011*). Alanine substitutions of three critical amino acids from the catalytic loop (D418), $Mg^{2+}$ binding loop (D436), or P+1 loop (T459) abolished the phosphorylation activity of NFRe (*Figure 1F*, lanes 4–12) showing that conserved residues from NFRe kinase are critical for its biochemical activity. Together, our results from biochemical studies demonstrate that *Nfre* encodes a LysM receptor kinase that can perceive Nod factor and has an active kinase.

### NFRe induces epidermal *Nin* expression

Since we now know that NFRe is an active LysM receptor with properties comparable to NFR1 in these in vitro assays, we next investigated the signalling capacities of NFRe compared to NFR1 in *Lotus* roots. We tested this by expressing NFRe in the *nfr1-1* mutant line containing the symbiotic *Nin*:GUS reporter (*Radutoiu et al., 2003*). Activation of the *Nin* promoter in *Nfre* transformed roots of *nfr1-1-Nin:GUS* plants, or the development of nodule and/or infection threads would indicate activation of symbiotic signalling. While *nfr1-1* roots transformed with the *Nfr1* gene developed *bona fide* root nodules and induced *Nin* promoter, those transformed with the empty vector, and thus expressing the native *Nfre* gene did not show any responses to inoculation with rhizobia (*Table 1* and *Figure 2—figure supplement 1*). These results indicate that *Nfre*, in its native status cannot replace the functions of NFR1. On the other hand, p35S-*Nfre* led to strong activation of the *Nin* promoter after inoculation with *M. loti* (*Figure 2A* and *Figure 2—figure supplement 1*). This symbiotic induction was however, only detected in the outer root layers (*Figure 2A*), and it was not followed by formation of nodules or infection threads even after 5 weeks of exposure to *M. loti* (*Figure 2—figure supplement 1*). This differed from the p35S-*Nfr1*-mediated signalling that induced *Nin* expression in both epidermal and cortical cells (*Figure 2B*) and led to formation of infected nodules

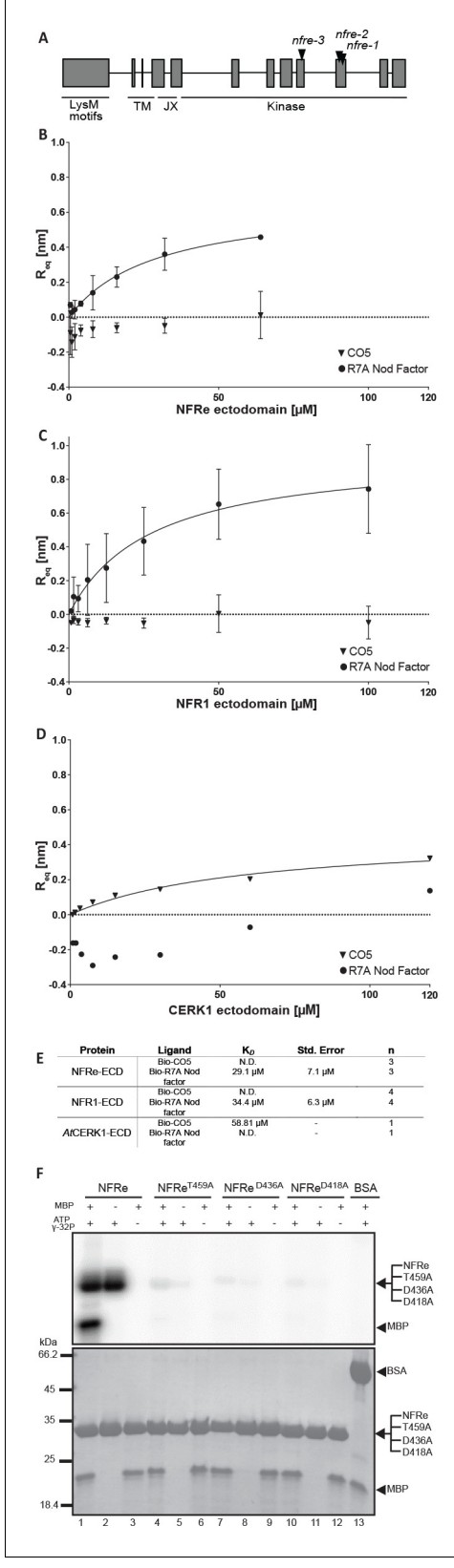

**Figure 1.** NFRe perceives Nod factor and has an active intracellular kinase. (**A**) The structure of *Nfre* gene (4663 bp) and predicted protein domains (600 aminoacids). *Figure 1 continued on next page*

(*Figure 2—figure supplement 1*). To understand whether this particular and cell layer specific activation of *Nin* by NFRe is a result of the expression of any LysM receptor of the NFR1-type, or specific to NFRe, *Nin* activation was assayed in *nfr1-1-Nin:GUS* plants transformed with the *Lys* paralogs of NFR1 (*Lohmann et al., 2010*). Under similar conditions, *Lys2*, *Cerk6* or *Lys7* driven by 35S promoter could not activate the *Nin* promoter, or induce nodule or infection thread formation in the *nfr1-1* mutant. (*Table 1* and *Figure 2—figure supplement 1*). These results demonstrate that in the presence of *M. loti*, NFRe, like NFR1, can initiate a symbiotic signalling cascade leading to *Nin* induction in *Lotus* roots, and that the cellular effects of this signalling are receptor-, and expression-dependent.

## NFRe maintains a low, epidermal expression during root nodule symbiosis

Previous studies based on transcript measurement showed that *Nfre* is expressed in *Lotus* roots (*Lohmann et al., 2010*). However, our results from the *nfr1-1* complementation studies revealed that expression of *Nfre* from p35S promoter is needed to induce observable *Nin* activation after rhizobia inoculation (*Figure 2—figure supplement 1*). To further understand the cause of this differential signalling we characterised the spatio-temporal regulation of *Nfre* in detail using GUS and tYFPnls (triple YFP-nuclear localised) reporter fusions, and measured the levels of *Nfre* transcript by quantitative RT-PCR. In uninoculated roots the *Nfre* promoter (2,6 kb) was preferentially active in root hair epidermal cells, in the susceptible zone of the root, and in the root tip (*Figure 2C,E*, and *Figure 2—figure supplement 2*). This differed from *Nfr1* that is expressed in the whole uninoculated root (*Radutoiu et al., 2003*; *Kawaharada et al., 2017*) (*Figure 2D*). The expression pattern of *Nfre* did not change after inoculation with *M. loti* (*Figure 2F* and *Figure 2—figure supplement 2*), indicating that, unlike *Nfr1* (*Radutoiu et al., 2003*; *Kawaharada et al., 2017*) and *Figure 2—figure supplement 2*), the expression of *Nfre* is not symbiotically regulated. Analyses of *Nfre* transcript levels in wild type roots either treated with Nod factor or inoculated with *M. loti* compared to control roots, further confirmed the unaltered expression observed from *Nfre* promoter studies (*Figure 2—figure supplement 2*). Direct comparison of *Nfr1* and *Nfre* transcript levels in uninoculated wild type roots showed a 3-fold higher level for *Nfr1*. Interestingly this difference was reduced

*Figure 1 continued*

The boxes indicate coding regions, lines are introns, and the location of mutations in the three alleles is indicated. The underlines indicate domains in NFRe; LysM domains, TM: transmembrane, JX: juxtamembrane, kinase. (B), (C) and (D) are binding curves obtained from the biolayer interferometry measurements of NFRe ectodomain, NFR1 ectodomain and *At*CERK1 ectodomain interaction with two different ligands, R7A Nod factor and GlcNAc5. Both NFRe and NFR1 ectodomain do not bind to GlcNAc5 but show binding to R7A Nod factor. *At*CERK1 ectodomain does not bind R7A Nod factor but binds GlcNAc5. (E) Binding constants of NFRe, NFR1 and *At*CERK1 ectodomain to GlcNAc5 and R7A Nod factor obtained from biolayer interferometry steady state-analysis. (F) *Nfre* encodes an active kinase domain. Autophosphorylation and protein kinase activities of wild-type NFRe, T459A, D436A, D418A NFRe mutant versions, and bovine serum albumin as control are shown. Myelin basic protein was used as substrate for kinase activities. Autoradiogram (top), and SDS-PAGE gels (bottom) are shown. K*D*, Std. Error and n represent the dissociation constant, standard deviation and number of biological replicates used for the analysis. N.D. represents not detectable.

DOI: https://doi.org/10.7554/eLife.33506.003

The following figure supplements are available for figure 1:

**Figure supplement 1.** NFRe is an NFR1 type LysM receptor.

DOI: https://doi.org/10.7554/eLife.33506.004

**Figure supplement 2.** Purification of NFRe, NFR1 and AtCERK1 ectodomains.

DOI: https://doi.org/10.7554/eLife.33506.005

**Figure supplement 3.** Chemoselective synthesis of biotinylated R7A Nod factor and chitin pentamer conjugates.

DOI: https://doi.org/10.7554/eLife.33506.006

significantly after Nod factor treatment (8 hr post treatment) or *M. loti* inoculation (2 and 3 dpi post inoculation), where *Nfr1* expression was down regulated, while *Nfre* maintained a low, but constant level (*Figure 2—figure supplement 2*). In summary, *Nfre* and *Nfr1* differ in their expression level and pattern in uninoculated roots, and follow a differential regulation during root nodule symbiosis. These differences could therefore, at least in part, account for the differential signalling capacities of the two LysM receptors.

## NFRe promotes nodule organogenesis

NFRe has the capacity to bind Nod factors in vitro (*Figure 1B*) and to induce a symbiotic signalling *in planta* when expressed in the *nfr1-1* mutant from the 35S promoter (*Figure 2A*). This prompted us to ask whether NFRe plays a role in root nodule symbiosis. Homozygous mutant plants from three independent alleles with exonic insertion of LORE1 retroelement (*Mun et al., 2016*) (*Figure 1A* and *Supplementary file 1*) formed significantly fewer nodules compared to wild type when grown in a binary association with *M. loti* (*Figure 3A*). The contribution of NFRe to root nodule organogenesis became even more evident when wild type and *nfre* mutants were grown in soil and were exposed to the native bacterial community. After 9 weeks, *nfre* mutants developed only half the number of wild type nodules (*Figure 3—figure supplement 1*). The shoot biomass and the general plant fitness were significantly reduced (*Figure 3—figure supplement 1*). Wild type plants had well-developed green pods, while *nfre* mutants had only few open flowers (*Figure 3—*

**Table 1.** *Nfre* expression from p35S promoter activates *Nin* induction in the *nfr1-1-Nin*:GUS plants

| Construct (plants analysed) | No. of plants *Nin* positive | % of plants with *Nin* induction | No. of nodulated plants | % of nodulated plants |
|---|---|---|---|---|
| Empty vector (28) | 0 | 0 | 0 | 0 |
| *pNfr1:Nfre* (21) | 0 | 0 | 0 | 0 |
| *p35S:Nfre* (58) | 28 | 48 | 0 | 0 |
| *p35S:Nfre_T459A* (21) | 0 | 0 | 0 | 0 |
| *pNfr1:Nfr1* (19) | 19 | 100 | 19 | 100 |
| *p35S:Nfr1* (26) | 25 | 96 | 25 | 96 |
| *p35S:Lys2* (14) | 0 | 0 | 0 | 0 |
| *p35S:Lys6* (34) | 0 | 0 | 0 | 0 |
| *p35S:Lys7* (34) | 0 | 0 | 0 | 0 |

DOI: https://doi.org/10.7554/eLife.33506.010

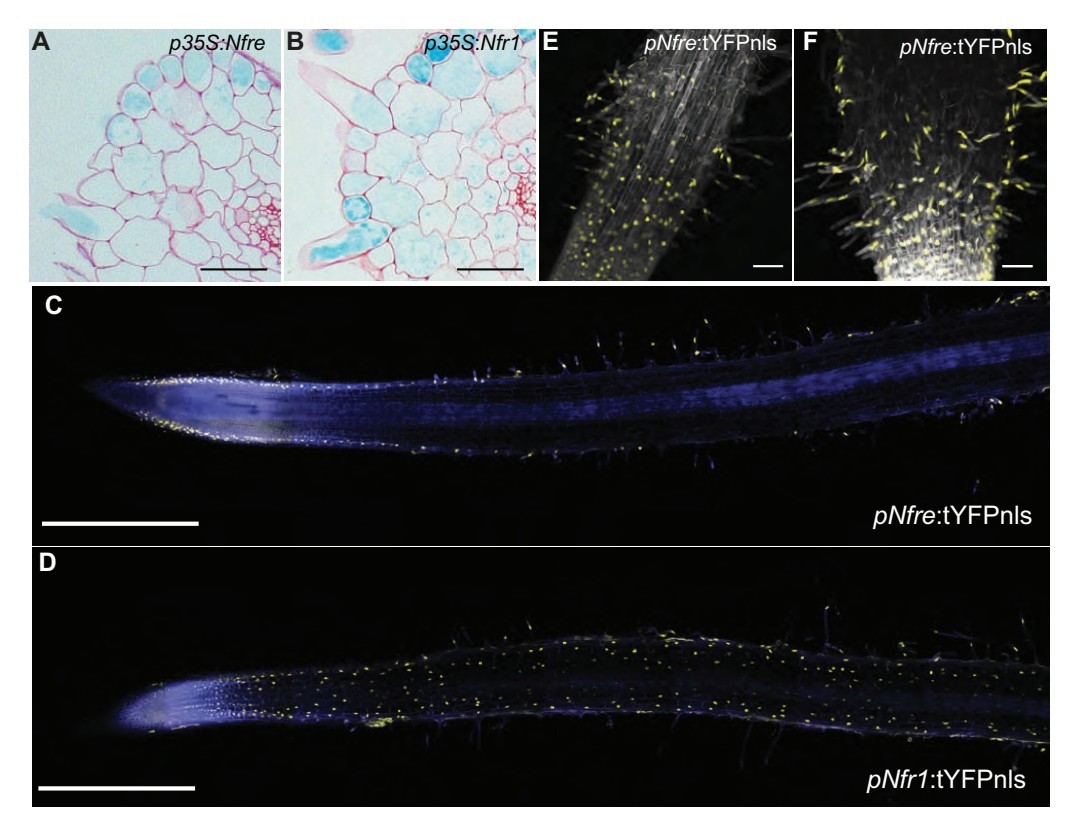

**Figure 2.** NFRe maintains a low, epidermal expression during root nodule symbiosis. (**A**) Transversal root section of *nfr1-1-Nin:*GUS plants expressing p35S-*Nfre* shows activation of *Nin* promoter in the outer cell layer after *M. loti* inoculation. (**B**) Transversal root section of *nfr1-1-Nin:*GUS plants expressing p35S-*Nfr1* shows activation of *Nin* promoter in all cell layers after *M. loti* inoculation. (**C**) The epidermal cells, primarily localized in the root susceptible zone, show activity of the *Nfre* promoter visualized by the nuclear localized triple YFP protein (tYFPnls). (**D**) Widespread activity of the *Nfr1* promoter in the uninoculated root visualized by nuclear localized triple YFP protein (tYFPnls). (**E**) The expression of *Nfre* in the susceptible zone of the root, and in the root hairs is maintained after inoculation with *M. loti* (**F**). Scale bars, 40 µm (**A, B**), 0.5 mm (**C, D**), and 50 µm (**E, F**).

DOI: https://doi.org/10.7554/eLife.33506.007

The following figure supplements are available for figure 2:

**Figure supplement 1.** Nin:GUS activation in *nfr1-1-Nin:GUS* plants expressing different receptor variants.
DOI: https://doi.org/10.7554/eLife.33506.008

**Figure supplement 2.** Expression patterns of LysM receptors in *Lotus japonicus.*
DOI: https://doi.org/10.7554/eLife.33506.009

*figure supplement 1*). Analyses of the dynamics of nodule primordia formation on plate-grown plants, revealed that *nfre* mutants, besides a noticeable reduced nodulation at the early time point (two wpi), had a significantly lower ability to reinitiate nodule formation on the expanding root system (five wpi) (*Figure 3B*). Unlike nodule organogenesis, the formation of infection threads (IT) appeared not to be affected by mutations in the *Nfre*. A similar number of ITs were present in wild type and *nfre* root hairs at 9 or 14 dpi (*Figure 3C*). The mature nodules formed on *nfre* appeared normally infected (*Figure 3—figure supplement 1*), and the proportion of pink/total nodules formed by soil-grown wild type and *nfre* plants was similar (*Figure 3—figure supplement 1*), indicating a normal infection process in the *nfre* mutants.

To further investigate the role of NFRe in Nod factor signalling we analysed its requirement for induction and maintenance of nuclear-associated calcium oscillations (spiking) after Nod factor treatment. Root hairs of wild type (n = 50) and *nfre-1* (n = 46) stable transgenics expressing the nuclear localised YC3.6 (Yellow Cameleon) showed clear signs of calcium oscillations after *M. loti* Nod factor treatment (*Figure 3D*) (app. 80% of the analysed cells responded). Closer inspection of the spiking frequency revealed that the average inter-spike interval was significantly longer in the *nfre* cells (106

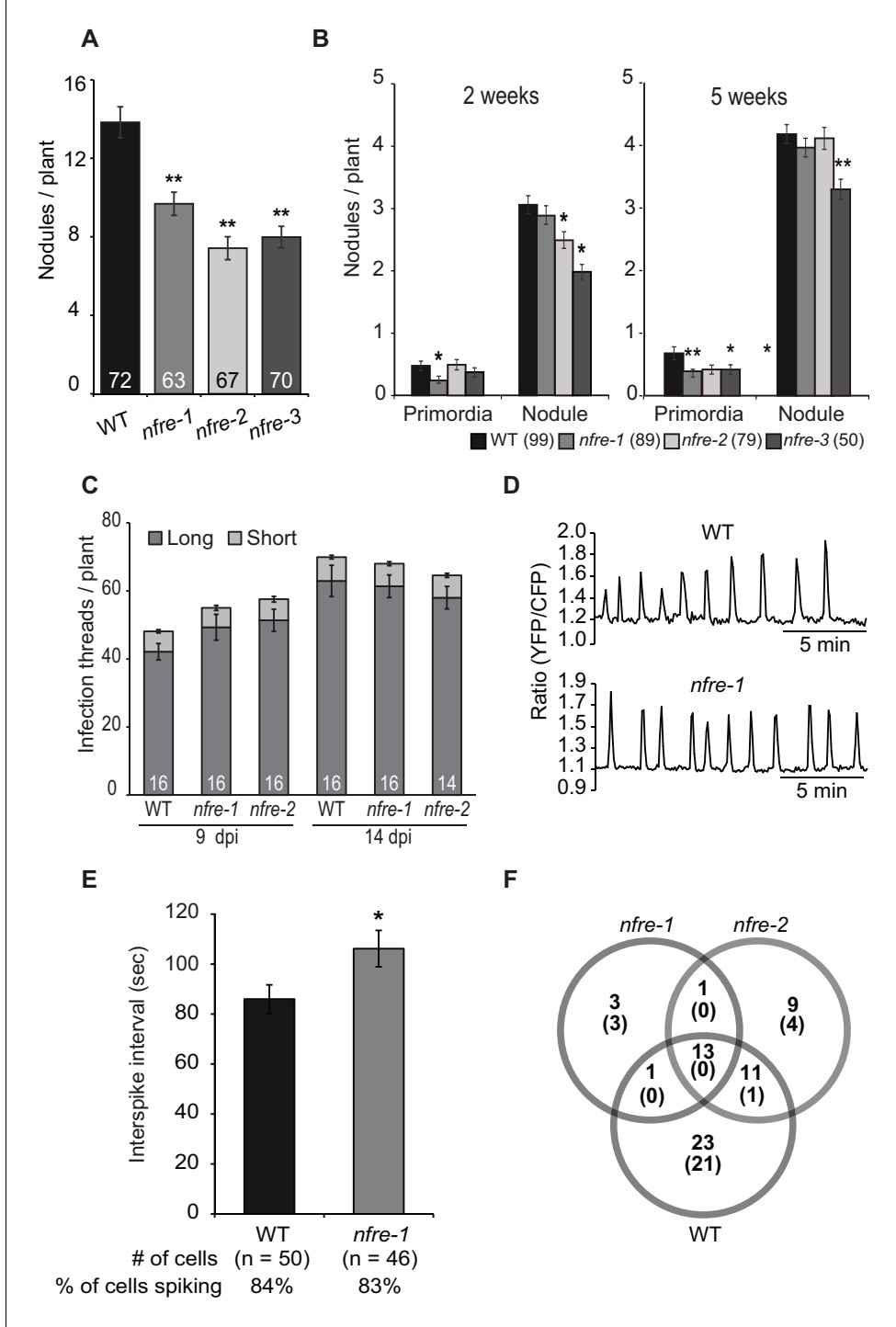

**Figure 3.** NFRe promotes nodule organogenesis in *Lotus japonicus*. (**A**) Greenhouse-grown *nfre* plants formed fewer root nodules compared to WT when analysed at eight wpi with *M. loti*. (**B**) Agar plate-grown *nfre* plants form fewer primordia than WT at 5 wpi with *M. loti*. (**C**) The *nfre* mutants and wild type plants form similar number of short and long root hair infection threads at 9 and 14 dpi. (**D**) Representative nuclear calcium oscillations (spiking) induced by R7A Nod factor ($10^{-8}$ M) in wild type and *nfre* mutant root hairs. $Ca^{2+}$ oscillations are presented as ratiometric values between YFP and CFP signals detected on the basis of the NLS-YC3.6 $Ca^{2+}$ sensor. (**E**) The inter-spike interval of *nfre-1* mutant is significantly longer than that of WT. (**F**) Venn diagrams of Nod factor up- and down-regulated (parentheses) genes detected in the susceptible zone at 24 hr after treatment. The values
*Figure 3 continued on next page*

*Figure 3 continued*

given at the bottom of columns in (**A**) and (**C**) represents the number of plants analysed. Error bars represent standard error of the mean. *p<0.05 and **p<0.01, Student's t-test compared to wild type.

DOI: https://doi.org/10.7554/eLife.33506.011

The following figure supplements are available for figure 3:

**Figure supplement 1.** The phenotype of wild-type and *nfre* plants.

DOI: https://doi.org/10.7554/eLife.33506.012

**Figure supplement 2.** Pattern of nuclear calcium oscillations in wild-type and *nfre-1* mutant.

DOI: https://doi.org/10.7554/eLife.33506.013

**Figure supplement 3.** Transcript levels of selected genes measured by quantitative RT-PCR in Mock, or Nod factor-treated wild type, *nfr1-1* and *nfre* mutant roots.

DOI: https://doi.org/10.7554/eLife.33506.014

**Figure supplement 4.** Chitin oligomers elicit production of similar ROS levels in *nfre* and wild type roots.

DOI: https://doi.org/10.7554/eLife.33506.015

s) compared to wild type (86 s), indicating that NFRe contributes to a constant interval length of calcium oscillations (*Figure 3—figure supplement 2*).

Next, we used RNA sequencing to investigate the requirement for *Nfre* in the transcriptional changes induced by *M. loti* Nod factor in the susceptible zone of the root at 24 hr after treatment. Genes that were differentially expressed (DEGs adjusted p<0.05) (Materials and methods) in Nod factor treated roots compared to water control were identified in wild type, *nfre-1* and *nfre-2* mutants. A large proportion of these (44 out of 90), which includes *Nin*, expansins, nodulins, receptors, transporters and transcription factors, were regulated by Nod factor in wild type but not in the *nfre* roots, indicating that their appropriate regulation in the susceptible zone, at 24 hr after Nod factor treatment is dependent on an active NFRe (*Figure 3—figure supplement 3* and *Supplementary file 2*). Other symbiosis-related genes like NFY-A, subtilase, and two genes encoding the cytokinin-induced message were found among the 13 DEGs in wild type and *nfre* mutants. Only one gene (an expansin) was found regulated by the Nod factor in both *nfre* mutants but not in wild type.

Our biochemical in vitro data based on BLI measurements shows that the NFRe ectodomain does not bind chitopentaose, suggesting that NFRe might not be involved in chitin signalling. To test this hypothesis *in planta* we measured the induction of reactive oxygen species (ROS) in response to CO8 or CO4 in the *nfre* mutants and wild type. We found that wild type, *nfre-1* and *nfre-2* mutants produced comparable levels of ROS, indicating that NFRe is unlikely to be involved in chitin signalling (*Figure 3—figure supplement 4*).

Together, these results show that NFRe represents an influential component of the epidermal Nod factor signalling in *L. japonicus*, promoting intracellular signalling that leads to optimal calcium spiking, activation of gene transcription and efficient nodule organogenesis on the expanding root system.

## The activation segments of NFR1 and NFRe determine the signalling output

The clear difference observed between NFR1 and NFRe in their ability to induce Nod factor signalling and spatial activation of the *Nin* promoter in *M. loti* inoculated *nfr1-1-Nin:GUS* roots prompted

**Table 2.** The intracellular domains of NFRe and NFR1 kinases determine the signalling output

| Construct (plants analysed) | No. of plants *Nin* positive | % of plants with *Nin* induction | No. of nodulated plants | % of nodulated plants |
|---|---|---|---|---|
| *p35S:NeK* (29) | 3 | 10 | 0 | 0 |
| *pLjUbi:NeK* (40) | 2 | 5 | 0 | 0 |
| *pLjUbi:NeKA1* (40) | 26 | 65 | 14 | 35 |
| *pLjUbi:Nfr1* (22) | 22 | 100 | 21 | 95 |

DOI: https://doi.org/10.7554/eLife.33506.016

us to identify the molecular determinants for this differential regulation. A chimeric receptor (NeK) containing the NFR1 extracellular domain followed by the transmembrane and intracellular kinase regions of NFRe (NFR1 ectodomain-NFRe kinase- NeK) (*Figure 1—figure supplement 1*) was constructed. This receptor was expressed in *nfr1-1-Nin:GUS* to test its capacity to induce activation of *Nin* promoter after *M. loti* inoculation. We observed that the signalling capacity of NeK receptor was similar to that of the NFRe, namely only epidermal induction of the *Nin* promoter (*Table 2* and *Figure 2—figure supplement 1*). This provides evidence for the presence of a molecular determinant for specific *Nin* induction in the intracellular regions of NFR1 and NFRe receptors. Alignment of the two kinases identified several divergent regions (*Figure 1—figure supplement 1*), but clear differences were found in the activation segment (*Figure 1—figure supplement 1*). Based on these differences, and previous knowledge (*Nakagawa et al., 2011*) that this region is crucial for kinase signalling and substrate recognition, we hypothesised that a specific NFR1/NFRe activation segment determines the specificity of the downstream signalling. We tested this hypothesis by swapping the NFRe activation segment with the corresponding region of NFR1 in the NeK receptor (NFR1 ectodomain-NFRe kinase with the Activation segment of NFR1 -NeKA1) (*Figure 1—figure supplement 1*). In contrast to the NeK receptor that induced *Nin* in the outer root cell layers of the *nfr1-1* mutant, NeKA1 led to cortical activation of *Nin* and nodule formation (*Table 2* and *Figure 2—figure supplement 1*).

These results show that the activation segment in NFR1 and NFRe determines the downstream signalling output in *Lotus* roots after *M. loti* inoculation.

## NFRe signalling is dependent of NFR5

Genetic and molecular studies established that a concerted NFR1-NFR5 signalling induces the nitrogen-fixing symbiosis (*Radutoiu et al., 2003*; *2007*). Here, we present evidence that in *L. japonicus* NFRe assists the development of root nodule symbiosis. Therefore, we hypothesised that NFRe-dependent signalling also involves NFR5. For this, we investigated the biochemical capacity of the NFRe kinase to transphosphorylate the intracellular NFR5 pseudokinase. The in vitro kinase assays showed that NFR5 is a substrate for the NFRe kinase. (*Figure 4A*, lanes 1–3) and that this transphosphorylation was dependent on the activation segment of the NFRe kinase. Mutation of T459 to A abolished the kinase activity of NFRe, while exchanging the native segment with the corresponding region of NFR1 maintained transphosphorylation (*Figure 4A*, lanes 4–6, 7–9). These results corroborate the observed nodule formation in the *nfr1-1* expressing the NeKA1 receptor (*Table 2* and *Figure 2—figure supplement 1*).

Next, we analysed the localisation and molecular properties of full-length NFRe and NFR5 using heterologous expression in *Nicotiana benthamiana*. The YFP tagged NFRe protein was found to localize to the plasma membrane and to co-localise with the plasma membrane marker, AtPIP2, like previously observed for NFR1 and NFR5 (*Madsen et al., 2011*) (*Figure 4—figure supplement 1*). Bimolecular fluorescence complementation (BiFC) analyses based on split YFP revealed that NFRe formed homomeric complexes alone and heteromeric complexes when co-expressed with either NFR1 or NFR5 (*Figure 4B* and *Figure 4—figure supplement 1*). The formation of heteromeric complex with NFR5 was not affected by kinase inactivation (*Figure 4B*). Like in the case of NFR1-NFR5 co-expression (*Madsen et al., 2011*), a signalling cascade leading to leaf cell death, dependent on an active NFRe kinase, was identified in *N. benthamiana* leaves co-expressing NFRe and NFR5 (*Figure 4—figure supplement 1*). Finally, we analysed whether the NFRe-dependent activation of *Nin* in *L. japonicus* roots was dependent on NFR5. *Nfre* driven by 35S promoter failed to induce *Nin*:GUS symbiotic reporter in the *nfr5-2* mutant background (*Figure 4—figure supplement 1* and *Supplementary file 3*). These results demonstrate that NFRe can interact with, and trans-phosphorylates NFR5 kinase, and induce a signalling cascade dependent on the NFR5 receptor.

Collectively, our results from biochemical studies of the extracellular and intracellular domains of NFRe, together with those obtained from mutant and functional analyses provide evidences for the involvement of NFRe ensuring a robust signalling for symbiosis with nitrogen-fixing rhizobia.

## Discussion

Nod factor binding by NFR1-NFR5 LysM receptors is required to induce nodule organogenesis and infection thread formation in *L. japonicus* (*Radutoiu et al., 2003*; *2007*). Here, we show that the

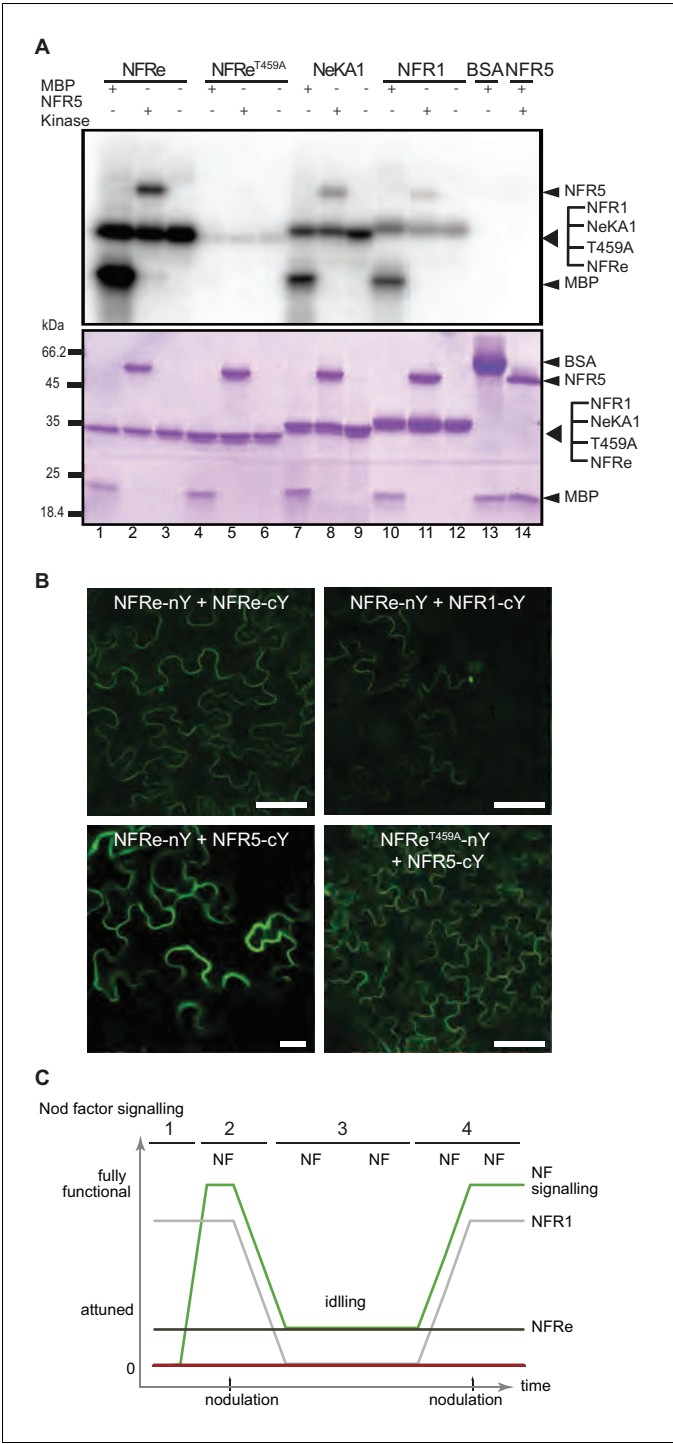

**Figure 4.** NFRe signalling is dependent of NFR5. (**A**) The NFRe kinase phosphorylates NFR5 kinase, whereas the NFReT459A shows no phosphorylation activity. The kinase of NeKA1 receptor in which the activation segment of NFRe was swapped with the corresponding region of NFR1 also phosphorylates NFR5 kinase. NFR1 kinase serves as positive control for NFR5 kinase transphosphorylation. Bovine serum albumin and NFR5 kinase domain are negative controls. (**B**) Bimolecular fluorescence complementation (BiFC) of YFP signal indicates protein-protein interactions in tobacco leaves. NFRe forms homomers (NFRe-nY +NFRe cY), and heteromers with NFR1 (NFRe-nY +NFR1 cY), or NFR5 (NFRe-cY +NFR5 nY). Formation of heteromeric complexes with NFR5 is not dependent on an active NFRe kinase (NFRe T459A -nY + NFR5 cY). (**C**) Working model of Nod factor signalling (green line) in the susceptible zone ensuring an efficient nodulation on the expanding root system. NFRe (dark grey line) has a constant expression in the epidermal cells of the susceptible zone. NFR1 (light grey line) dominates the

*Figure 4 continued on next page*

*Figure 4 continued*
uninoculated root in terms of expression level and spatial distribution (1). Once the symbiotic process is initiated by the Nod factor (NF), the expression of NFR1 is rapidly downscaled (2). A sustained expression of NFRe in the epidermal cells ensures an idling signalling in the susceptible zone, keeping the expanding root system tuned in to rhizobia (3). NFR1 acts as a master switch triggering recurrent symbiotic events in a fast and efficient manner from NFRe-attuned epidermal cells (4).
DOI: https://doi.org/10.7554/eLife.33506.017
The following figure supplement is available for figure 4:

**Figure supplement 1.** NFRe is localized on plasma membrane and signals together with NFR5.
DOI: https://doi.org/10.7554/eLife.33506.018

NFR1-NFR5 signalling cascade operates on the framework provided by the epidermal LysM receptor NFRe. NFRe and NFR1 share biochemical and molecular properties that is similar Nod factor-binding affinity, and chitopentaose differentiating capacity when assessed by biolayer interferometry (*Broghammer et al., 2012*), functional kinases dependent on fully operative domains (*Madsen et al., 2011*), capacity to phosphorylate, and induce a signalling cascade dependent of NFR5 (*Madsen et al., 2011*). In spite of these similarities, NFR1 and NFRe have evolved distinct biological properties defined by specific spatio-temporal expression, and downstream signalling cascades controlled by diverged kinases.

The epidermis of the expanding root system is continuously exposed to Nod factors produced by rhizobia present in the rhizosphere. Nevertheless, the number and the location of primordia guiding the epidermal infection threads are precisely determined. Complex regulatory networks involving transcriptional regulators, hormones, shoot- and root-derived signals (*Ferguson et al., 2010*; *Miyata et al., 2013*; *Sasaki et al., 2014*; *Miri et al., 2016*; *Roy et al., 2017*), as well as tightly controlled receptor signalling (*Mbengue et al., 2010*; *Kawaharada et al., 2017*), collaborate to coordinate how many nodules the plant develops. With this framework in mind, a working model is emerging when considering our findings (*Figure 4C*). This model incorporates the interplay of the NFR1 and NFRe in the epidermis, ensuring efficient and robust signalling in the susceptible zone of the expanding root system. In the absence of the symbiont, *Nfre* has a low and constant expression in the susceptible zone, while *Nfr1* outnumbers *Nfre* in terms of expression level and spatial distribution (*Figure 4C-1*). Once the symbiotic process is initiated, the expression of NFR1 is rapidly downscaled in the susceptible zone (*Figure 4C-2*). A sustained expression of NFRe in the epidermal cells of the susceptible zone could ensure an idling signalling, keeping the expanding root system tuned in to rhizobia (*Figure 4C-3*). NFR1 acts as a master switch triggering recurrent symbiotic events in a fast and efficient manner from NFRe-attuned epidermal cells (*Figure 4C–4*).

In general, protein-carbohydrate interactions are usually weak and low-affine (micromolar-millimolar range) (*Holgersson et al., 2005*) and signalling therefore, emerges as being controlled by ligand multivalency and/or by receptor multiplicity (*Kiessling and Pohl, 1996*; *Rabinovich, 2002*; *Vasta et al., 2012*). In line with this notion studies of receptors present at the plant and mammalian plasma membrane revealed a conserved strategy to ensure specific, instantaneous, switchable and evolvable downstream signalling; namely, increased responsiveness and specificity via combinatorial systems (*Ostrom et al., 2001*; *Piñeyro, 2009*; *Bodmann et al., 2015*; *Bücherl et al., 2017*). The signalling properties of NFRe remain to be determined, but our findings based on the properties of this LysM receptor kinase, together with the symbiotic phenotypes of *nfre* mutants unveil a more complex signalling operating in the epidermal cells of *L. japonicus* than anticipated from studies of the basic and essential receptor-components.

It is possible that multiple LysM receptors assemble into functional signalling complexes where signalling specificity is the result of the nature of the complex, rather than isolated LysM receptors alone. The mechanistic details of NFR1-NFRe signalling remain to be discovered, but we envision that differences might exist among legumes, since *nfr1* in *Lotus* and *lyk3* or *sym37* in *Medicago* and pea have different symbiotic phenotypes (*Radutoiu et al., 2003*; *Smit et al., 2007*; *Zhukov et al., 2008*), indicating distinct evolutionary trajectories after separation of the IRLC (Inverted Repeat-lacking clade) legumes (*Sprent, 2008*). Tandem NFR1-type receptors are found in all legumes and in non-legume species as well (*Liang et al., 2013*; *De Mita et al., 2014*). Ample comparative

phylogenomics and trans-complementation studies targeting tandem duplicated LysM receptors will greatly help determining their evolutionary impact and their role in different plant species.

## Materials and methods

### Phylogenetic tree and alignment

Clustal Omega was used to prepare multiple sequence alignment for phylogenetic analysis. The region between 55 and 251 in this alignment was realignment to adjust the positions of CXC motif. This alignment was used for the phylogenetic analysis with Neighbor Joining. The distance was measured with Jukes-Cantor and the bootstrap was 1000 replicates. These alignment and phylogenetic analyses were performed in the CLC Main Workbench v7.9.1. The amino acid sequence of OsCERK1 (Os08g0538300-01) is available in The Rice Annotation Project Data Base (rap-db). The other sequences below are available in NCBI: AtCERK1 (NP_566689), NFR1 (CAE02590), NFRe (AB503681), LYS2 (AB503682), EPR3 (AB503683), LYS4 (AB503685), LYS5 (AB503686), LYS6 (AB503687), LYS7 (AB503688).

### Expression and purification of NFRe, NFR1 and AtCERK1 ectodomains

NFRe and NFR1 ectodomain boundaries were defined by secondary structure prediction performed with PSIPRED (*Buchan et al., 2013*). Their signal peptides were predicted using the SignalP 4.1 server (*Petersen et al., 2011*). The AtCERK1 ectodomain boundaries were designed based on the reported crystal structure (*Liu et al., 2012*). The predicted ectodomain sequences were codon-optimized for insect cell expression and synthesized with an N-terminal gp67 secretion signal peptide and a c-terminal hexa-histidine tag (GenScript, Piscataway, USA) and inserted into the pOET4 transfer vector (Oxford Expression Technologies). Recombinant AcMNPV baculoviruses were produced in Sf9 cells cultured with SFX (Hyclone) or TNM-FH medium (Sigma-Aldrich) supplemented with 10% (v/v) FBS (Gibco), 1% (v/v) chemically defined lipid concentrate (Gibco) and 1% (v/v) Pen/Strep (10,000 U/ml, Life Technologies). The FlashBac Gold kit (Oxford Expression Technologies) was utilized according to the manufacturer's instructions. Viruses were amplified until a third passage virus culture of 500 mL was obtained. For large scale protein expression Sf9 cells were infected with 5% (v/v) of the passage three virus solution and cultured in suspension with serum-free SFX insect cell medium (Hyclone) or BD BaculoGold MAX-XP medium (BD Biosciences, discontinued) supplemented with chemically defined lipid concentrate and Pen/Strep as described above. The culture was maintained in a shaking incubator at 26°C for five days, after which the medium was harvested by centrifugation in a Sorvall RC5plus centrifuge (SLA-1500 rotor) at 6000 rpm at room temperature for 25 min. Subsequently, the cleared medium was dialyzed against 10 volumes of buffer A (50 mM Tris-HCl pH 8, 200 mM NaCl) for one day at 4°C with the buffer being exchanged at least four times. The proteins were loaded on a HisTrap excel column (GE Healthcare) equilibrated with buffer A and recirculated over 3 days at 4°C using a peristaltic pump. After a washing step with buffer W (50 mM Tris-HCl pH 8, 500 mM NaCl and 20 mM imidazole) proteins were eluted with buffer B (50 mM Tris-HCl pH 8, 200 mM NaCl, 500 mM imidazole). Imidazole was removed by dialyzing against buffer A and the purity was improved by a second IMAC purification step using a HisTrap HP column (GE Healthcare). The NFRe ectodomain was dialyzed against a low salt buffer (50 mM Tris-HCl and 50 mM NaCl) before purification on a MonoQ column (GE Healthcare). The resulting flow-through containing NFRe was collected and concentrated in a Vivaspin column (10 kDa cut-off, Sartorius Stedim biotech). NFRe and NFR1 were finally purified by size exclusion chromatography using a Superdex 200 10/300 GL column (GE Healthcare) and AtCERK1 using a Superdex 75 10/300 GL size exclusion column (GE Healthcare) in SEC buffer (Phosphate buffered saline, pH 7.2, 500 mM NaCl). At each purification step, yield and purity were assayed by SDS-PAGE.

### Synthesis of biotinylated R7A Nod factor and chitopentaose conjugates

Biotin conjugates were synthesized using a two-step procedure according to *Figure 1—figure supplement 3*. O-(2-Aminoethyl)-N-methyl hydroxylamine trifluoroacetic acid salt was prepared as described previously (*Bohorov et al., 2006*), and all other chemicals were purchased from Sigma-Aldrich and used without further purification. Nod factors from *Mesorhizobium loti*, strain R7A, NodMl-V(C18:1Δ11Z, Cb, Me, AcFuc), containing three main species (3-O-acetylated, 4-O-

acetylated, or non-acetylated fucosyl unit) were purified as described previously (*Bek et al., 2010*). Purified R7A Nod factor (3.6 mg, 2.29 μmol, 5 mM) was dissolved in 0.62 M NaOAc buffer, pH 4.5, containing 50% acetonitrile, and *O*-(2-aminoethyl)-*N*-methyl hydroxylamine trifluoroacetic acid salt (150 mM, 30 equiv.) was added. The resulting mixture was allowed to react at room temperature for 16 hr, after which it was concentrated under a nitrogen flow. The intermediate product was purified by semipreparative HPLC on an UltiMate 3000 instrument fitted with a Waters 996 photodiode detector, using a Phenomenex Luna 5 μm, C18(2), 100 Å, 250 × 100 mm semi-preparative column. An isocratic elution at 40% acetonitrile in water, 5 mL/min for 30 min was used. The intermediate eluted at 9.5–11.5 min. Conjugate formation was confirmed by HR-MS (ES+): calcd for [M + H, 1Ac] + = 1573.8081, found 1573.8185. The purified intermediate was dissolved at a concentration of 10 mM in 50 mM sodium tetraborate buffer, pH 8.5, containing 50% acetonitrile. NHS-dPEG4-biotin (15 mM, 1.5 equiv.) was added. The resulting mixture was allowed to react at room temperature for 16 hr. The biotin conjugate product was purified by semipreparative HPLC as described above. A gradient of 5–100% acetonitrile in water over 40 min, running at 5 mL/min, was used. The conjugate eluted after 21 min (68% acetonitrile). The chromatogram displayed a broad product peak due to the presence of the three species differing in substitution on the fucosyl residue. The biotin-R7A Nod factor conjugate (18% yield) was quantified using the HABA/avidin biotin quantification kit (Pierce). HR-MS (ES+): calcd for [M + 2 hr, 1Ac]2 += 1024.0175, found 1024.0184, and calcd for [M + 2 hr, 0Ac]2 += 1003.0122, found 1003.0129 (*Figure 1—figure supplement 3*). High-resolution mass spectra (HR-MS) were obtained using a Dionex Ultimate 3000 UHPLC instrument (Thermo) coupled to a Bruker Impact HDII QTOF mass spectrometer. The synthesis of a biotin-chitopentaose (GlcNAc)five conjugate was performed essentially as for the biotin-R7A Nod factor conjugate. The product was purified by HPLC using a Phenomenex Luna 5 μm, C18(2), 100 Å, 250 × 100 mm semi-preparative column, using a gradient of 5–100% acetonitrile in water, 5 mL/min for 40 min. The product eluted after 12.7 min (35% acetonitrile). The final yield of the biotin-(GlcNAc)five conjugate was determined to be 9%. HR-MS (ES+): calcd for [M + 2 hr]2 += 790.3552, found 790.3557 (*Figure 1—figure supplement 3*).

## Biolayer interferometry

Binding of NFRe, NFR1 and AtCERK1 ectodomains to biotin-R7A Nod factor and biotin-(GlcNAc)5 (CO5) was measured on an Octet RED biolayer interferometer (Pall ForteBio). Biotinylated R7A Nod factor and (GlcNAc)5, were immobilized on streptavidin biosensors (for kinetics, Pall ForteBio) at a concentration of 250 nM for 5 min. Immobilization levels of biotin-R7A and biotin-(GlcNAc)five were followed during immobilization and amounted to approximately 2.4 nm and 0.4 nm of saturation, respectively. Interaction with NFRe, NFR1 and AtCERK1 ectodomains was measured in dilution series at protein concentrations ranging from 0.5 to 64 μM (NFRe), 0.78–100 μM (NFR1) or 0.93–160 μM (AtCERK1) for 10 min. Subsequently, dissociation was recorded for 5 min. All steps were conducted in phosphate-buffered saline, pH 7.4, 500 mM NaCl, 0.01% Tween20. Parallel background measurements using biosensors immobilized with free biotin were subtracted from R7A Nod factor and (GlcNAc)five curves to correct for unspecific binding. Sensorgrams were processed using Forte-Bio Data Analysis 7.0 (Pall ForteBio). Equilibrium dissociation constants from steady-state analysis were calculated in GraphPad Prism 6 (GraphPad Software) by nonlinear regression using the response at equilibrium (Req) plotted against protein concentration.

## NFRe kinase domain expression and purification

The NFRe, NFR1 and NFR5 kinase domains were predicted using TMHMM Server v. 2.0 and PSIPRED secondary structure prediction. NFRe kinase domain was cloned into pET-30 Ek/LIC vector (Novagen), NFR1 and NFR5 kinase domains were cloned into pET-32 Ek/LIC vectors (Novagen). Three NFRe kinase domain mutants were created using the Quikchange Lightning Site-Directed Mutagenesis Kit (Agilent Technologies) according to the manufacturers' instructions. For the chimeric kinase of NeKA1, cDNA fragment was assembled by PCR as described previously (*Heckman and Pease, 2007*). NFRe kinase was expressed into *E. coli* BL21-CodonPlus, NFR1 and NFR5 kinase into *E. coli* Rosetta 2. Cultures were grown until OD600 = 0.8 and cold-shocked for 30 min in an ice bath. Protein expression was subsequently induced with 1 mM IPTG and left overnight to shake at 20°C. Cells were harvested by centrifugation at 3300 rpm in a Sigma swing-out rotor 13855 and afterwards

resuspended in 100 ml lysis buffer (50 mM Tris-HCl pH8, 400 mM NaCl, 1 mM Benzamidine, 20 mM Imidazole, 5 mMβ-mercaptoethanol and 10% (v/v) glycerol). Resuspended pellets were broken by sonication and cell debris removed by centrifugation at 12000 rpm (F21S-8 × 50 y rotor, Thermo-Fisher). The resulting supernatant was loaded on a Ni-NTA IMAC affinity column (ThermoFisher) equilibrated with lysis buffer at 4°C using a peristaltic pump. After a wash step with buffer W-kinase (50 mM Tris-HCl pH 8, 1 M NaCl, 1 mM Benzamidine, 50 mM Imidazole, 5 mM β-mercaptoethanol and 10% Glycerol) to remove contaminants, kinases were eluted with buffer B-kinase (50 mM Tris-HCl pH 8, 300 mM NaCl, 500 mM imidazole, 5 mM β-mercaptoethanol and 10% Glycerol). His-tagged TEV protease (homemade) was added to the eluted proteins at a 1:100 (w:w) ratio and dia-lysed against a dialysis buffer (50 mM Tris-HCl pH 8, 300 mM NaCl, 5 mM β-mercaptoethanol and 10% Glycerol) overnight at 4°C. The cleaved kinase domain proteins were subjected to a second round of IMAC affinity column purification and collected in the flow-through. The kinase domain pro-teins were concentrated in a Vivaspin column (10 kDa cut-off) before being subjected to size exclu-sion chromatography using either a Superdex 75 or Superdex 200 10/300 GL columns on ÄKTA Purifier system (both GE Healthcare). Purification was performed by isocratic elution in Buffer GF (50 mM Tris-HCl pH 8, 300 mM NaCl and 5 mM β-mercaptoethanol). After each purification step, yield and purity were assayed using SDS-PAGE.

## In vitro kinase assay

4 µg of purified kinase domain proteins were incubated in Kinase Activity Buffer (2 mM MnCl2, 2 mM NaCl, 2 mM MgCl2, 1 mM ZnCl2, 50 mM HEPES pH 7 and 100 µM ATP) and 2mCi ATP, [gamma-32P] (PerkinElmer) in a total reaction volume of 20 µL. 2 µg Myelin Basic Protein (Sigma Aldrich) and 4 µg NFR5 kinase domain were added to the appropriate reactions. Additionally, con-trols without Myelin Basic Protein, ATP [gamma-32P] were made. The reactions were left to incubate for 1 hr at room temperature before loading and running on a 15% SDS-PAGE Gel. After staining with Coomassie Brilliant Blue, the gel was transferred on a phosphor plate and exposed overnight before scanning on a Typhoon Scanner (GE Healthcare).

## Plant material

*Lotus japonicus*, ecotype B–129 Gifu (*Handberg and Stougaard, 1992*) is the wild type used for all experiments. Homozygous *nfre* (previously called *lys1*) mutants were identified in the LORE1 collec-tion (*Fukai et al., 2012*; *Urbański et al., 2012*) and the primers used for genotyping are listed in *Supplementary file 1*. Seeds were sterilized and germinated and the 3 days old seedlings were transferred to the corresponding conditions below.

## Bacterial strains and constructs

*Mesorhizobium loti*, strain R7A labelled with GFP or dsRed, and NZP2235 were used for phenotypic analyses. An inoculum density of OD600 = 0.02 was used for all studies. *Agrobacterium rhizogenes* AR1193 (*Stougaard et al., 1987*) was used for hairy root transformation. *A. tumefasiens* AGL1 was used for the transient expression in leaves of *N. benthamiana*.

The various constructs used for *L. japonicus* transformation were assembled using Golden Gate Cloning (*Engler et al., 2014*), and constructs for *N. benthamiana* were assembled using Gateway system with 35S promoter driving the expression. The details of each construct and primers for clon-ing are presented in *Supplementary file 1*. All constructs were confirmed by sequencing.

## Complementation and promoter analysis using hairy root

The seedlings for hairy root transformation were moved to half-strength B5 media and transformed as described previously (*Stougaard, 1995*). The composite plants were transferred to Magenta boxes containing sterile clay granule substrate or to sterile agar plates supplemented with ¼ B and D media and inoculated.

## GUS staining and cross section

Transformed roots were incubated in GUS staining buffer [0.5 mg/mL X-Gluc, 50 mM phosphate buffer (pH7.0), 5% methanol, 1 mM K4(Fe(CN)6), 1 mM K3(Fe(CN)6), 0.05% Triton X-100] at 37°C for 18 hr in dark. The samples were washed with 50 mM phosphate buffer (pH7.0) and stored in 70%

ethanol at 4°C. GUS stained roots and nodules were observed using a Leica M165FC stereomicroscope and Leica DFC 310 FX camera system. Three to five representative samples were used for generating transversal sections, as described previously (*Gavrilovic et al., 2016*).

## Microscopic observations for promoter analysis using tYFPnls

For promoter activity using tYFPnls, transformed roots were fixed with paraformaldehyde and cleared as described previously (*Warner et al., 2014*). The samples were analysed on a ZEISS confocal microscope LSM780. The whole root images were obtained using Z-stack and tail scan tools, the images of root surface were obtained using Z-stack tool. Final images were generated by Maximum Intensity Projection in ZEN software (ZEISS) or ImageJ.

## Quantitative RT-PCR

For transcript measurement by quantitative RT-PCR, 3 days seedlings were moved to agar plates supplemented with 1/4 B and D media and whole roots of 12 days-old (*Figure 2—figure supplement 2*) or 14 days old (*Figure 3—figure supplement 3*) plants were harvested after specific treatment as specified in each experiment. The mRNA was isolated from whole roots (*Figure 2—figure supplement 2*) or the susceptible zone (*Figure 3—figure supplement 3*) using Dynabeads mRNA DIRECT TM kit (Invitrogen). RevertAid Reverse Transcriptase (Fermentas) was used for cDNA synthesis. The quantitative RT-PCR was performed with LightCycler 480 II and LightCycler 480 SYBR Green I Master mix (Roche). ATP-synthase (ATP), Ubiquitin-conjugating enzyme (UBC) and Protein phosphatase 2A (PP2A) were used as reference genes. The three biological (each consisting of 40 plants in *Figure 2—figure supplement 2* or 30 plants in *Figure 3—figure supplement 3*) and three technical repetitions were used to calculate the geometric mean of the relative transcript levels and the corresponding upper and lower 95% confidence.

## Plant phenotyping

Sterile agar plates or clay granule substrate supplemented with ¼ B and D media was used for phenotypic analysis in laboratory and greenhouse conditions (*Figure 3A,B*). Cologne soil (*Zgadzaj et al., 2016*) with no additional nutrients or inoculum was used for plant phenotyping in *Figure 3—figure supplement 1*.

## Observation of infection threads (IT)

The 3 day old seedlings were transferred to agar plates supplemented with ¼ B and D media. After 3 days growth, the plants were inoculated with *M. loti* R7A labelled with dsRed. The infection threads were counted at 9 and 14 days after inoculation.

## Nuclear calcium oscillation in root hairs

Seedlings were grown on agar plates supplemented with 1/4 B and D with 12.5 µg/mL AVG for 1–2 weeks. One seedling was transferred to a glass slide and Nod factor treatment was performed using $10^{-8}$ M *M. loti* R7A Nod factor solution. The samples were analysed on a confocal microscope LSM780 (ZEISS) and a water lens (W plan-Apochromat 40x/1.0 DIC M27, ZEISS). YC3.6 was excited at 458 nm, and emissions from ECFP and cpVenus were split into different detectors and collected at 463 to 509 and 519 to 621 nm. Calcium spiking was monitored for up to 3 hr after the Nod factor treatment on each root. Several regions of the same root were monitored for 10 to 30 min, and minimum five nuclei were monitored on each root. In total, 50 nuclei from wild type and 46 nuclei from *nfre-1* root hairs were monitored. The fluorescence intensity data collected in the first 10 min for each nucleus was analysed by CaSA software (*Russo et al., 2013*). For calculation to the mean time between $Ca^{2+}$ spikes (inter spike interval, ISI) for each genotype, the mean of ISI for one cell was used.

## RNA sequencing

For RNA sequencing, 3 days seedlings were moved to agar plates supplemented with 1/4 B and D media and susceptible zone of 14 days-old plants was harvested after specific treatment as specified in *Figure 3F*. The total RNA was isolated from the susceptible zone (15 mm root pieces) using Nucleo spin RNA plant (Macherey-Nagel). Total RNA (>0.8 µg) from two biological replicas per

sample was used by GATC Biotech (Germany) to prepare random primed cDNA library and for sequencing with Illumina HiSeq: read length $1 \times 50$ bp. For the analysis of the RNA sequencing data the read trimming and mapping were performed by CLC genomics workbench 9.5.3 using *Lotus japonicus* v3.0 at Lotus base (https://lotus.au.dk/) (*Mun et al., 2016*), as reference. Differentially expressed genes (log2 fold change >0 or<0, adjusted *p* value < 0,05) were determined using the DESeq2 R package, with the 'fittype' parameter set to 'local' and the 'betaprior' parameter to 'true'. The HTS filter R package was integrated in the DESeq2 pipeline before calling for differentially expressed genes, in order to remove from the analysis the genes with low read counts. Venn diagrams were generated with the VennDiagram R package (*Chen and Boutros, 2011*).

### Reactive oxygen species (ROS) measurements

Seedlings were germinated and grown on a stack of wet filter paper in upright position at 21°C under 16/8 hr light/dark conditions. Roots of 7 day old seedlings were cut to 0.5 cm pieces, collected to white 96 well flat bottom polystyrene plates (Greiner Bio-One) and kept overnight in sterile water in darkness at room temperature to recover from stress before the treatment. ROS measurements were conducted in a Varioskan Flash Multimode Reader (Thermo Scientific) in luminometric measurement mode. The reaction mixture consisted of the respective elicitor, 20 µM luminol (Sigma) and 5 µg/ml horseradish peroxidase (Sigma). As elicitor 1 µM tetra-N-acetyl-chitotetraose, CO4 (Megazyme) or octa-N-acetyl-chitooctaose, CO8 (IsoSep) was used. In the negative control wells water was replacing the elicitor. In one measuring well six roots (10 mg root material) was used. In one repetition three wells were measured for every treatment for every genotype. At least two repetitions were conducted with similar results.

### Protein localization and BIFC studies in *N. benthamiana* leaves

*N. benthamiana*, infiltrated leaves were analysed after 3 days using a Zeiss LSM510 MetaConfocal microscope. The leaves were infiltrated with 0.8 M mannitol to induce plasmolysis. The samples were mounted in 30% glycerol on the slide. For cell death in *N. benthamiana*, infiltrated leaves were observed after 4 days.

### Data availability

RNA-seq reads were deposited at ArrayExpress (accession: E-MTAB-5855).

## Acknowledgements

We would like to thank Rebecca Fitchett for proofreading the manuscript, Terry Mun for computational analysis support, Mette Hofmann Asmussen for technical assistance and Finn Pedersen for taking care of the plants. This work was supported by the Danish National Research Foundation grant no. DNRF79, and by The Bill and Melinda Gates Foundation, as part of Engineering the Nitrogen Symbiosis for Africa.

## Additional information

### Funding

| Funder | Grant reference number | Author |
|---|---|---|
| Danmarks Grundforsknings-fond | DNRF 79 | Jeryl Cheng<br>Kira Gysel<br>Zoltan Bozsoki<br>Yasuyuki Kawaharada<br>Christian Toftegaard Hjuler<br>Kasper Kildegaard Sørensen<br>Simon Kelly<br>Mikkel Boas Thygesen<br>Maria Vinther<br>Knud Jørgen Jensen<br>Michael Blaise<br>Lene Heegaard Madsen<br>Kasper Røjkjær Andersen |

| Engineering Nitrogen Symbio-sis for Africa | Engineering Nitrogen Symbiosis for Africa | Eiichi Murakami Noor de Jong Jens Stougaard Simona Radutoiu |
|---|---|---|

The funders had no role in study design, data collection and interpretation, or the decision to submit the work for publication.

### Author contributions

Eiichi Murakami, Data curation, Formal analysis, Supervision, Validation, Investigation, Visualization, Methodology, Writing—original draft, Writing—review and editing; Jeryl Cheng, Data curation, Formal analysis, Validation, Investigation, Visualization, Methodology, Writing—original draft, Writing—review and editing; Kira Gysel, Data curation, Formal analysis, Supervision, Validation, Investigation, Visualization, Writing—original draft, Writing—review and editing; Zoltan Bozsoki, Data curation, Formal analysis, Investigation, Writing—original draft; Yasuyuki Kawaharada, Christian Toftegaard Hjuler, Kasper Kildegaard Sørensen, Data curation, Investigation, Methodology; Ke Tao, Noor de Jong, Maria Vinther, Dorthe Bødker Jensen, Formal analysis, Investigation, Methodology; Simon Kelly, Data curation, Formal analysis, Investigation, Methodology; Francesco Venice, Data curation, Validation, Investigation, Visualization, Methodology; Andrea Genre, Formal analysis, Validation, Visualization, Methodology; Mikkel Boas Thygesen, Formal analysis, Validation, Visualization, Methodology, Writing—original draft, Writing—review and editing; Knud Jørgen Jensen, Michael Blaise, Formal analysis, Supervision, Methodology; Lene Heegaard Madsen, Formal analysis, Supervision, Investigation, Visualization, Methodology; Kasper Røjkjær Andersen, Data curation, Formal analysis, Supervision, Validation, Visualization, Methodology, Writing—original draft, Writing—review and editing; Jens Stougaard, Funding acquisition, Writing—original draft, Writing—review and editing; Simona Radutoiu, Conceptualization, Supervision, Funding acquisition, Visualization, Writing—original draft, Project administration, Writing—review and editing

### Author ORCIDs

Eiichi Murakami https://orcid.org/0000-0001-6414-0254
Zoltan Bozsoki https://orcid.org/0000-0002-4267-9969
Christian Toftegaard Hjuler https://orcid.org/0000-0001-6055-6391
Andrea Genre https://orcid.org/0000-0001-5029-6194
Mikkel Boas Thygesen https://orcid.org/0000-0002-0158-2802
Kasper Røjkjær Andersen https://orcid.org/0000-0002-4415-8067
Jens Stougaard https://orcid.org/0000-0002-9312-2685
Simona Radutoiu https://orcid.org/0000-0002-8841-1415

### Decision letter and Author response

Decision letter https://doi.org/10.7554/eLife.33506.026
Author response https://doi.org/10.7554/eLife.33506.027

## Additional files

### Supplementary files

• Supplementary file 1. Primers used in this study.
DOI: https://doi.org/10.7554/eLife.33506.019

• Supplementary file 2. Differentially expressed genes in the susceptible zone of wild-type and nfre mutant roots after Nod factor treatment.
DOI: https://doi.org/10.7554/eLife.33506.020

• Supplementary file 3. Signalling from NFRe in *Lotus* roots is dependent of NFR5.
DOI: https://doi.org/10.7554/eLife.33506.021

• Transparent reporting form
DOI: https://doi.org/10.7554/eLife.33506.022

## Data availability

Sequencing data have been deposited under code: E-MTAB-5855

The following dataset was generated:

| Author(s) | Year | Dataset title | Dataset URL | Database, license, and accessibility information |
|---|---|---|---|---|
| Murakami | 2017 | Lotus japonicus response to Nod factor: wild type and nfre mutants | https://www.ebi.ac.uk/ar-rayexpress/experiments/E-MTAB-5855/ | Publicly available at the European Nucleotide Archive (accession no: E-MTAB-5855) |

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
