## [Decision Letter]

[Editors’ note: this article was originally rejected after discussions between the reviewers, but the authors were invited to resubmit after an appeal against the decision.]

Thank you for submitting your work entitled "Epidermal Nod factor receptor NFRe ensures robust symbiotic signalling in *Lotus japonicus* roots" for consideration by *eLife*. Your article has been reviewed bythree3 peer reviewers, and the evaluation has been overseen by a Reviewing Editor and a Senior Editor. The reviewers have opted to remain anonymous.

Our decision has been reached after consultation between the reviewers. Based on these discussions and the individual reviews below, we regret to inform you that your work will not be considered further for publication in *eLife*.

The identification of eNFR as an 'idling state' epidermis-specific Nod factor receptor in *Lotus japonicus* is an important and novel finding. In particular, eNFR-mediated differential spatio-temporal signaling in partnership with NFR5 bears the potential of re-shaping the field's view on Nod factor perception and signaling. However, as substantial concerns about the ligand specificity of this newly identified receptor and its predominant epidermal localization remain we are unable to recommend publication of your manuscript in its present form.

Reviewer #1:

Simona Radutoiu and Jens Stougaard have shown over the last decades that the LysM receptor kinase gene family has a crucial role in the perception of rhizobial signaling and exopolysaccharides. The gene family is large and rapidly evolving, making the analysis of the function of individual members challenging, further impeded by partial redundancy and limited functional conservation between model legume species. The manuscript by Murakami Eiichi et al., describes another LysM-domain containing receptor-like kinase (NFRe) that is closely related to the Nod factor receptor NFR1. Like previously shown for NFR1, the presently described NFRe binds Nod-factor and plays a role in the symbiotic signalling in *Lotus japonicus*. The authors confirm the predicted Nod-factor binding of the NFRe ectodomain by bio-layer interferometry with a biotinylated Nod factor and demonstrate that the kinase kinase domain is active with an in vitro kinase assay. Because of the biochemical properties and the sequence conservation with the known RLK NFR1, the authors conclude a close relationship of the two proteins. The expression of NFRe is restricted to epidermal cells and the expression pattern does not change after Nod-factor treatment. They show this with a promoter-GUS assay and promoter-fluorophore reporter fusion in microscopic pictures. The nfre mutant exhibits a much weaker nodulation phenotype (reduced nodule number) compared to the nfr1 or nfr5 mutants, which are completely unable to nodulate. The authors also claim that nfre mutants shows altered calcium spiking after inoculation which they measured using a calcium sensitive FRET reporter. According to the authors the transcriptome determined via RNA sequencing is also altered after inoculation with rhizobia. In addition, they suggest but do not demonstrate, an interaction of NFRe with NFR5. In the discussion, they propose that NFRe plays a role in attuning epidermal cells for rhizobial infection. Overall the work employs a wide variety of experimental techniques.

There are a number of major issues that must be resolved in order to properly support some of their major claims (see below).

Importantly, the authors fail to make clear what aspects of the identification and characterization of NFRe constitute a level of scientific novelty sufficient to justify publication in *eLife*.

1) The authors claim that nfre mutants show altered calcium spiking. However, in figure 3—figure supplement 3B,C,D it is not apparent how many samples were taken. If I understood correctly from the material and methods, 5 nuclei were measured. This is sample size is too small for a student’s t-test. The nodulation specific pattern of calcium spiking may differ between cells inside and outside the susceptible zone. The selection of the nuclei may therefore be especially relevant and must be comparable between the genotypes analyzed. Also, the potential influence of background mutations must be ruled out. It is perfectly fine if the authors drop this claim in case they cannot provide sufficient statistical support for it.

2) The authors claim that NFRe specifically binds Nod-factors and with a similar Kd as NFR1. This is supported by a bio-layer interferometry binding assay. However, considering that the Radutoiu/Stougaard laboratories in Aarhus have already claimed differential binding competence of individual LysM receptors for at least three different ligands, the specificity of NFRe must be supported with a necessary array of controls available in their laboratories. Here, they only included CO5 as a negative control. Because of the potential binding competence of LysM domains for exopolysaccharide fragments as shown for the putative exopolysaccharide LysM receptor EPR3 (Kawaharada et al., 2015) the inclusion of this ligand appears essential. Ideally, the authors also include as a negative control the EPR3 ectodomain to demonstrate the specificity of their assay and of the NFRe ectodomain.

3) The authors claim that NFRe is a paralog of NFR1 but shows a distinct downstream signalling based on differences in the kinase activation segment. However, they switch between three different promoters and use inconsistent sets of control and swap constructs. In particular the key domain swap called N1_E.1 lacks the NFR1 and NFRe control constructs. This makes it impossible to reach conclusions from the quantitative complementation results. The NFR1 promoter constructs lack the N1_E.1 swap construct, the ubiquitin promoter constructs lack the NFR1 and NFRe controls and the 35S constructs do not contain the N1_E.1 swap. The experimental strategy underlying this construct arrangement is obscure.

4) The authors claim that nfre mutants show reduced nodulation. Mutants were analysed two, five and eight weeks post inoculation and all the mutants are carrying 'significantly' lower nodules only eight weeks post inoculation. However, in the relevant figure 3—figure supplement 1, the statistical test applied (student’s t-test) is inappropriate for the small subset of individual plants investigated.

5) The authors claim that NFRe does not undergo expression changes after inoculation with Nod factor or rhizobia. However, the expression level of NFRe and NFR1 after treatment with Nod-factor or Rhizobia were measured by qRT-PCR shown in figure 2—figure supplement 3. It compares the expression levels to a mock treated plant. The mock treatment is not specified also not in the Material and methods section. The mock treatment is only shown for one of the three timepoints measured, therefore the variation in expression without treatment is not clear.

6) The authors claim that NFRe interacts with NFR5, based on a BiFC assay in *N. benthamiana* leaves. The limited validity of this assay for membrane bound proteins is well recognized in the community. One reason is the inability to provide negative controls because of the high non-specific interaction frequency resulting from overexpression and the diffusion of membrane proteins in only two dimensions. Consistent with this problem, the authors of this work also fail to provide a proper negative control.

7) The authors claim that NFRe phosphorylates NFR5. However, the kinase assay lacks controls demonstrating that the phosphorylation of NFR5 by NFRe is specific. Because plant receptor kinase kinase domains are highly promiscuous with regards to their phosphorylation substrate in in vitro kinase assays, the results do not allow the conclusion that the kinase interacts with NFR5 in vivo.

8) The authors claim that the presence of NFRe in the epidermal cells keeps them tuned to rhizobia also after downregulation of NFR1 during symbiotic signaling. This is the conclusion of the claims mentioned above for which some need further investigation.

Reviewer #2:

Nod factor signaling is initiated by LysM domain receptor kinases and in Lotus NFR1 and NFR5 play essential roles. As a result of their strong phenotypes, there is general acceptance in the symbiosis field that the nod factor receptors are known (NFR1/NFR5) and therefore the other LysM domain kinases must be involved in other aspects and not in nod factor signaling. As shown here, this is not true and complete nod factor signaling is achieved through different receptor complexes. In this careful and comprehensive study of NFRe, the authors reveal the contribution of a second receptor kinase and show that it is necessary for the complete signaling required to attain wildtype levels of nodulation. NFRe contributes by providing different spatio-temporal signaling and does so through partnership with NFR5. These findings are unexpected but that data are convincing (with one caveat mentioned below) and demonstrate the contribution of NFRe which, in *Lotus japonicus*, plays a minor role relative to the dominant receptor NFR1. Not only does the study illustrate complexities in nod factor signaling, it is very likely that it will provide the basis for answers to differences observed in nodulation signaling among the legume where currently significant differences in the phenotypes of the receptor mutants are not understood. The NFR1/NFRe duplication occurs in all legumes and different levels of contribution from the two paralogs may explain many of the differences between legumes. These data will cause the symbiosis field to reconsider nod factor signaling.

One of the major claims is a difference in spatial and temporal expression patterns of the two receptors. While the Q-RT-PCR and promoter-GUS data are convincing, I found it difficult to see differences between NFR1 and NFRe promoter-YFP-NLS (Figure 2). Maybe additional panels could be added to illustrate the expression differences in the susceptible zone.

Reviewer #3:

In this paper, the authors report that an epidermal LysM receptor kinase (LYS1/NFRe) perceives Nod factors and contributes to rhizobial symbiosis, probably by keeping the epidermal cells at an "idling" state. Although the involvement of NFRe in the symbiotic signaling is evident from the results with the corresponding mutants but the detailed mechanism by which NFRe contributes to the symbiosis is still largely obscure. Importantly, biochemical characterization of NFRe contains several problems that raise serious questions on the "receptor" function of NFRe.

The authors analyzed the interaction of the heterologously expressed ectodomains of NFRe and NFR1 using bilayer interferometry, where the biotinylated derivatives of a Nod factor and (GlcNAc)5 were immobilized onto the sensor tip. The results are quite strange. Potato lectin, which is known to bind chitin oligosaccharides but nothing to do with symbiosis, showed a very high affinity (Kd, 30 nM) to the Nod factor, while both NFRe and NFR1 showed thousand times lower (!) affinities to the Nod factor. The affinities of NFRe and NFR1 to the Nod factor (Kd, approximately 30 μM) are also far low compared to the biological activity of Nod factors. In general, the instrument used here seems to give a rather high estimate of the affinity, because the affinity of potato lectin to (GlcNAc)5 obtained here (Kd, 5.2 nM) seems to be much higher than the traditional estimate of the affinity of this lectin to chitin oligosaccharides (The lectins, Academic Press). Thus, it is quite questionable whether such a low affinity of NFRe and NFR1 to the Nod factor observed in this work reflects a biologically significant interaction. I am afraid that such a very low affinity simply reflects some non-specific interaction of these proteins with the hydrophobic region of Nod factor, fatty acid moiety.

To avoid such a possibility, the authors should incorporate inhibition type experiments in their binding studies. If the observed binding of NFRe and NFR1 to the Nod factor reflects a biologically significant interaction, it should be inhibited by the biologically active Nod factor but not by inactive Nod factors or chitin oligosaccharides.

There are some problems for the characterization of the expressed proteins too. Although the authors described "Proteins of high purity were obtained after four steps of purification, as confirmed by the homogeneous and sharp peaks observed after size exclusion chromatography" (Subsection “NFRe is a Nod factor receptor with an active intracellular kinase”), the results in Figure 1—figure supplement 2B shows the presence of multiple peaks for NFR1-ECD. If the authors used only the peak indicated as NFR1-ECD, they should describe it in the figure legend. It is better to show the elution profiles of final preparations of NFRe/NFR1-ECD by using the same column. The authors should also explain why these preparations showed multiple bands in SDS-PAGE (Figure 1—figure supplement 2C). It is also questionable why the sizes of the smallest bands for NFR1-ECD and NFRe-ECD are very different, while the expected sizes of these proteins are quite similar. I am afraid that some limited proteolysis happened for NFR1-ECD.

Characterization of the nfre mutants clearly showed that NFRe contributes to symbiotic signaling and nodule formation. However, the results of the overexpression experiments seem to be difficult to understand. In the nfr1 background, slight increase of the copy number of Nfre gene resulted nothing but the overexpression of the same gene induced a limited symbiotic response induced by *M. loti*, i.e., activation of Nin promotor in the epidermal cell layer. This Nin activation was not associated with the formation of nodules. Is it safe to connect the results only observed with the overexpression experiments to physiological function? Does the result fit to the hypothesis of "idling" function of NFRe?

The Venn diagram shown in Figure 3F indicates the presence of significant differences between nfre-1 and nfre-2. Is it OK?

Would it be better to confirm the results of nfre mutants by complementation?

[Editors’ note: what now follows is the decision letter after the authors submitted for further consideration.]

Thank you for choosing to send your work entitled "Epidermal Nod factor receptor NFRe ensures robust symbiotic signalling in *Lotus japonicus* roots" for consideration at *eLife*. Your article and your letter of appeal have been considered by a Senior Editor and the Reviewing Editor of your manuscript, and we regret to inform you that we are upholding our original decision.

Although we acknowledge the technical issue, it is difficult that a method hardly used in the field should explain obvious discrepancies noted by the referees. In this context, it also is somewhat problematic that if the reviewers do not catch it, the majority of readers won't either. Thus, the binding studies should be complemented with additional assays, using a more frequently used reliable standard method, such as SPR or MST. We would be ready to evaluate a resubmission of a manuscript that has been amended accordingly.

[Editors’ note: what now follows is the decision letter after the authors submitted for further consideration.]

Thank you for submitting your article "Epidermal Nod factor receptor NFRe ensures robust symbiotic signalling in *Lotus japonicus* roots" for consideration by *eLife*. Your article has been reviewed by three peer reviewers, and the evaluation has been overseen by a Reviewing Editor and Christian Hardtke as the Senior Editor. The reviewers have opted to remain anonymous.

The reviewers have discussed the reviews with one another and the Reviewing Editor has drafted this decision to help you prepare a revised submission.

Nod factor signaling is initiated by LysM domain receptor kinases and in Lotus NFR1 and NFR5 play essential roles. Your finding of NFRe, a receptor kinase that it is necessary for the complete signaling required to attain wildtype levels of nodulation, is intriguing and novel. In addition, the finding that NFRe contributes by providing different spatio-temporal signaling and does so through partnership with NFR5, is exciting and will impact future research on how plant rhizobia symbiosis is established.

While all referees appreciate much of the information reported they still request additional experimentation and clarification to be provided. These points are listed below:

1) The highly volatile nature of the spiking frequency makes it very difficult to determine fundamental differences. Thus, more detailed dissection of the patterns is required to support your conclusions. Likewise, strong statements in the text and the abstract should be softened in order to reflect the strength of the phenotype observed in the nfre1-1 mutant.

2) The referees believe that (transient) complementation of the nfre1-1 mutant is required to overcome problems associated with quantitatively differing phenotypes observed in the three nfre1-1 mutant alleles.

3) The use of inhibition type experiments is mandatory in addition to the differential binding experiments. This is important in order to demonstrate specificity of binding, which in addition to ligand affinity is a major criterion for biologically meaningful receptor-ligand interaction. Use of Nod factor, chitin oligosaccharide or exopolysaccharide as competitors in biolayer interferometry are likely to strengthen your claim about the receptor function of NFRe.

You may also wish to show that the biotinylated R7A Nod factor retains the biological activity.

[Editors’ note: what now follows is the decision letter after the authors submitted for further consideration.]

Thank you for submitting your article "Epidermal Nod factor receptor NFRe ensures robust symbiotic signalling in *Lotus japonicus* roots" for consideration by *eLife*. Your article has been reviewed by two peer reviewers, and the evaluation has been overseen by Thorsten Nürnberger as the Reviewing Editor and Christian Hardtke as the Senior Editor. The reviewers have opted to remain anonymous.

The reviewers have discussed the reviews with one another and the Reviewing Editor has drafted this decision to help you prepare a revised submission.

The referees acknowledge to a great extent the changes introduced into your manuscript. They also realize that you have tried hard to tackle the problem of demonstrating nod factor binding specificity to NFRe, which unfortunately was unsuccessful for technical reasons. Nevertheless, the identification of NFRe and genetic evidence for its role in nodulation remains a very valuable finding. We would therefore recommend publication of your study given you re-write those chapters of your manuscript that describe NFRe as a nod factor binding site (receptor). Instead, you may refer to eNFR as a protein involved in nodulation for which receptor function is yet to be shown. You may also briefly refer to technical problems you had with handling this protein in order to demonstrate its receptor function, which may be useful information for our readership. Likewise, you may state what type of experiment would be needed to demonstrate receptor specificity.

---

## [Author Response]

[Editors’ note: the author responses to the first round of peer review follow.]

The identification of eNFR as an 'idling state' epidermis-specific Nod factor receptor in Lotus japonicus is an important and novel finding. In particular, eNFR-mediated differential spatio-temporal signaling in partnership with NFR5 bears the potential of re-shaping the field's view on Nod factor perception and signaling. However, as substantial concerns about the ligand specificity of this newly identified receptor and its predominant epidermal localization remain we are unable to recommend publication of your manuscript in its present form.

We are happy to see that the reviewers and the editors appreciate the novelty and the high relevance of our findings. We have identified and addressed the concerns on NFRe capacity to bind Nod factor, and of chosen controls, have performed additional experiments and addressed reviewer’s comments in detail.

Reviewer #1:

Simona Radutoiu and Jens Stougaard have shown over the last decades that the LysM receptor kinase gene family has a crucial role in the perception of rhizobial signaling and exopolysaccharides. The gene family is large and rapidly evolving, making the analysis of the function of individual members challenging, further impeded by partial redundancy and limited functional conservation between model legume species. The manuscript by Murakami Eiichi et al. describes another LysM-domain containing receptor-like kinase (NFRe) that is closely related to the Nod factor receptor NFR1. Like previously shown for NFR1, the presently described NFRe binds Nod-factor and plays a role in the symbiotic signalling in Lotus japonicus. The authors confirm the predicted Nod-factor binding of the NFRe ectodomain by bio-layer interferometry with a biotinylated Nod factor and demonstrate that the kinase kinase domain is active with an in vitro kinase assay. Because of the biochemical properties and the sequence conservation with the known RLK NFR1, the authors conclude a close relationship of the two proteins. The expression of NFRe is restricted to epidermal cells and the expression pattern does not change after Nod-factor treatment. They show this with a promoter-GUS assay and promoter-fluorophore reporter fusion in microscopic pictures. The nfre mutant exhibits a much weaker nodulation phenotype (reduced nodule number) compared to the nfr1 or nfr5 mutants, which are completely unable to nodulate. The authors also claim that nfre mutants shows altered calcium spiking after inoculation which they measured using a calcium sensitive FRET reporter. According to the authors the transcriptome determined via RNA sequencing is also altered after inoculation with rhizobia. In addition, they suggest but do not demonstrate, an interaction of NFRe with NFR5. In the discussion, they propose that NFRe plays a role in attuning epidermal cells for rhizobial infection. Overall the work employs a wide variety of experimental techniques.

We thank reviewer #1 for the appreciative words on our work and efforts in defining the role of LysM receptors despite the inherent difficulties.

There are a number of major issues that must be resolved in order to properly support some of their major claims (see below).Importantly, the authors fail to make clear what aspects of the identification and characterization of NFRe constitute a level of scientific novelty sufficient to justify publication in eLife.

Since their discovery in 2003, NFR1/NFR5 have been considered the sole Nod factor receptors in *L. japonicus*. This is primarily because of their critical role in the initiation of nodule development and rhizobial infection. Our detailed characterisation of NFRe demonstrates that this capacity to perceive and signal after Nod factor perception is not solely mediated through the known NFR1/NFR5 receptors – and robust signalling is dependent on NFRe. Our results shear light on the complexity in Nod factor perception, that unlike previously thought, seems to require receptor multiplicity to ensure switchable and evolvable signalling.

To accommodate this major comment, we have adjusted our Discussion section that reads now: “In general, protein-carbohydrate interactionsare characterised by a low affinity^36^. Signalling therefore, emerges as being controlled by ligand multi-valency and/or by receptor multiplicity^37-39^. In line with this notion studies of receptors present at the plant and mammalian plasma membrane revealed a conserved strategy to ensure specific, instantaneous, switchable and evolvable downstream signalling; namely, increased responsiveness and specificity via combinatorial systems^40-43^. Our findings on the biochemical and molecular properties of NFRe, together with the symbiotic phenotypes of nfre mutants unveil an example of a more complex signalling operating in the epidermal cells of *L. japonicus* than anticipated from studies of the basic and essential receptor-components.”

1) The authors claim that nfre mutants show altered calcium spiking. However, in figure 3—figure supplement 3B,C,D it is not apparent how many samples were taken. If I understood correctly from the material and methods, 5 nuclei were measured. This is sample size is too small for a student’s t-test. The nodulation specific pattern of calcium spiking may differ between cells inside and outside the susceptible zone. The selection of the nuclei may therefore be especially relevant and must be comparable between the genotypes analyzed. Also, the potential influence of background mutations must be ruled out. It is perfectly fine if the authors drop this claim in case they cannot provide sufficient statistical support for it.

We would like to assure reviewer #1 that we did take into consideration the different regions of the root and that we mentioned this in the Materials and methods section: “Several regions of the same root were monitored for 10 to 30 min, …” Furthermore, in the main Figure 3 we present the actual number of nuclei analysed for calcium spiking in wild-type (n=50) and in nfre (n=46). To make this point clearer we have now added the numbers to the main text and to the methods section that reads now: “In total, 50 nuclei from wild type and 46 nuclei from nfre-1 root hairs were monitored.” For clarity, in the Figure 3—figure supplement 2 we present the pattern of calcium spiking in representative nuclei.

2) The authors claim that NFRe specifically binds Nod-factors and with a similar Kd as NFR1. This is supported by a bio-layer interferometry binding assay. However, considering that the Radutoiu/Stougaard laboratories in Aarhus have already claimed differential binding competence of individual LysM receptors for at least three different ligands, the specificity of NFRe must be supported with a necessary array of controls available in their laboratories. Here, they only included CO5 as a negative control. Because of the potential binding competence of LysM domains for exopolysaccharide fragments as shown for the putative exopolysaccharide LysM receptor EPR3 (Kawaharada et al., 2015) the inclusion of this ligand appears essential. Ideally, the authors also include as a negative control the EPR3 ectodomain to demonstrate the specificity of their assay and of the NFRe ectodomain.

Previous work to which we have contributed significantly demonstrated the capacity of LysM ectodomains to bind and signal after perception of specific carbohydrate ligands; Nod factor, COs, peptidoglycan, exopolysaccharide. This is not unusual for receptors as previously shown for the LRR ectodomains that are able to perceive e.g. a diverse panel of peptides. We agree that performing large comparative studies based on single ectodomains binding capacities might be informative, but these need to be interpreted in the context of their biological function. We now included AtCERK1 as control and demonstrate that NFRe binds Nod factor and discriminates COs in a similar way as NFR1, while AtCERK1 shows the opposite binding specificities. Furthermore, none of our results obtained from nfre mutant characterisation pointed towards a role in exopolysaccharide perception, which were shown to be important for infection thread formation and elongation. The nfre mutants have a normal rate of bacterial infection in the epidermis, and inside the nodules. Additionally, inoculation with the *M.loti* exoU mutant resulted in reduction of the number of nodule primordia like observed for *M.loti* wild-type, rather than rescue of the infection phenotype as observed for the epr3 mutants (Murakami et al., personal communication).

3) The authors claim that NFRe is a paralog of NFR1 but shows a distinct downstream signalling based on differences in the kinase activation segment. However, they switch between three different promoters and use inconsistent sets of control and swap constructs. In particular the key domain swap called N1_E.1 lacks the NFR1 and NFRe control constructs. This makes it impossible to reach conclusions from the quantitative complementation results. The NFR1 promoter constructs lack the N1_E.1 swap construct, the ubiquitin promoter constructs lack the NFR1 and NFRe controls and the 35S constructs do not contain the N1_E.1 swap. The experimental strategy underlying this construct arrangement is obscure.

We agree that our presentation did not distinguish the different aims of these experiments. The choice of promoter was driven by the scientific question to be addressed. First, we investigated the role of NFRe expression for its functionality by driving Nfre from Nfr1 and 35S promoters. This revealed that a high level of expression from 35S promoter was needed to drive Nin induction after Rhizobium inoculation. Next, we use this promoter and a highly expressed promoter of Lotus (Lj-ubiquitin) to drive the expression of chimeric constructs defining the role of different intracellular regions of NFR1 and NFRe. We have revised the manuscript (Results section) for clarity.

4) The authors claim that nfre mutants show reduced nodulation. Mutants were analysed two, five and eight weeks post inoculation and all the mutants are carrying 'significantly' lower nodules only eight weeks post inoculation. However, in the relevant figure 3—figure supplement 1, the statistical test applied (student’s t-test) is inappropriate for the small subset of individual plants investigated.

The reviewer #1 is right, the nfre mutants show a significant reduction in the number of nodules at 5 weeks when using the plate system or when analysed after 8 weeks in greenhouse or soil conditions. This apparent “late” but significant and robust response across three growth conditions demonstrates its role in ensuring a robust symbiotic signalling in the epidermis necessary for NFR1/NFR5 master switch that trigger recurrent symbiotic events on the growing root system. The statistical test for the Figure 3—figure supplement 1 was adapted to the number of analysed plants (Kruskal-Wallis) and a significant difference between wildtype and mutant plants was observed.

5) The authors claim that NFRe does not undergo expression changes after inoculation with Nod factor or rhizobia. However, the expression level of NFRe and NFR1 after treatment with Nod-factor or Rhizobia were measured by qRT-PCR shown in figure 2—figure supplement 3. It compares the expression levels to a mock treated plant. The mock treatment is not specified also not in the Material and methods section. The mock treatment is only shown for one of the three timepoints measured, therefore the variation in expression without treatment is not clear.

Numerous observations of uninoculated transgenic roots expressing Nfr1/Nfr5/Nfre-GUS or tYFP fusions revealed no detectable changes in the expression pattern or intensity, consequently we used their expression levels in roots at 8h after water treatment (Mock) as baseline.

6) The authors claim that NFRe interacts with NFR5, based on a BiFC assay in N. benthamiana leaves. The limited validity of this assay for membrane bound proteins is well recognized in the community. One reason is the inability to provide negative controls because of the high non-specific interaction frequency resulting from overexpression and the diffusion of membrane proteins in only two dimensions. Consistent with this problem, the authors of this work also fail to provide a proper negative control.

We show that NFRe can interact with NFR5 in *N. benthamiana*, that its kinase can phosphorylate NFR5 kinase and that Nin induction by 35S-NFRe in Lotus roots is dependent of NFR5. All together, we use three different experimental systems (in vitro, *N. benthamiana* leaves and Lotus roots) and show that NFRe can interact with NFR5 like it was previously shown for NFR1.

We do not understand what the reviewer means by lack of “proper negative control for this work”. We provide evidences for the lack of BiFC in the absence of the respective tested partners (Figure 4—figure supplement 1).

7) The authors claim that NFRe phosphorylates NFR5. However, the kinase assay lacks controls demonstrating that the phosphorylation of NFR5 by NFRe is specific. Because plant receptor kinase kinase domains are highly promiscuous with regards to their phosphorylation substrate in in vitro kinase assays, the results do not allow the conclusion that the kinase interacts with NFR5 in vivo.

We agree with the reviewer #1 that kinases tend to be highly promiscuous and considering the number of potential targets it is not possible to demonstrate specificity. We have therefore not claimed specificity for NFRe-NFR5 interaction. We show that NFRe kinase can phosphorylate NFR5 and we do not draw the conclusion of NFRe interaction with NFR5 on the basis of this trans-phosphorylation capacity alone. We have however, shown that the activation loops of NFR1 and NFRe are major determinants of the downstream signalling from the two kinases.

We refer the reviewer to the following sentence in subsection “NFRe signalling is dependent of NFR5”, “These results demonstrate that NFRe can interact with, and trans-phosphorylates NFR5 kinase, and induce a signalling cascade dependent on the NFR5 receptor. Collectively, our results from biochemical studies of the NFRe extracellular and intracellular domains, together with those obtained from mutant and functional analyses provide evidences for a novel epidermal Nod factor receptor ensuring a robust signalling for symbiosis with nitrogen-fixing rhizobia.” that clearly state the combined experimental evidence for NFRe function.

8) The authors claim that the presence of NFRe in the epidermal cells keeps them tuned to rhizobia also after downregulation of NFR1 during symbiotic signaling. This is the conclusion of the claims mentioned above for which some need further investigation.

We have addressed the above comments and revised the manuscript. The results obtained from nfre mutant analyses, functional studies of NFRe in the nfr1 mutant and biochemical evidences for Nod factor binding suggest a model for the interplay between NFR1 and NFRe on the expanding root system. We believe that identification of a new Nod factor receptor in legume roots with a differential function opens up for novel studies for fine dissection of the signalling initiated at the plasma membrane.

Reviewer #2:Nod factor signaling is initiated by LysM domain receptor kinases and in Lotus NFR1 and NFR5 play essential roles. As a result of their strong phenotypes, there is general acceptance in the symbiosis field that the nod factor receptors are known (NFR1/NFR5) and therefore the other LysM domain kinases must be involved in other aspects and not in nod factor signaling. As shown here, this is not true and complete nod factor signaling is achieved through different receptor complexes. In this careful and comprehensive study of NFRe, the authors reveal the contribution of a second receptor kinase and show that it is necessary for the complete signaling required to attain wildtype levels of nodulation. NFRe contributes by providing different spatio-temporal signaling and does so through partnership with NFR5. These findings are unexpected but that data are convincing (with one caveat mentioned below) and demonstrate the contribution of NFRe which, in Lotus japonicus, plays a minor role relative to the dominant receptor NFR1. Not only does the study illustrate complexities in nod factor signaling, it is very likely that it will provide the basis for answers to differences observed in nodulation signaling among the legume where currently significant differences in the phenotypes of the receptor mutants are not understood. The NFR1/NFRe duplication occurs in all legumes and different levels of contribution from the two paralogs may explain many of the differences between legumes. These data will cause the symbiosis field to reconsider nod factor signaling.

We thank reviewer #2 for pointing out the current status of understanding in the field and the relevance of our new findings.

One of the major claims is a difference in spatial and temporal expression patterns of the two receptors. While the Q-RT-PCR and promoter-GUS data are convincing, I found it difficult to see differences between NFR1 and NFRe promoter-YFP-NLS (Figure 2). Maybe additional panels could be added to illustrate the expression differences in the susceptible zone.

We have revised the manuscript and included additional results on the NFR1/NFRe expression pattern. Figure 2 contains now the expression pattern of the two receptors as observed from one optical section. This illustrates the epidermal expression of NFRe and the whole root expression of NFR1. Supplementary Figure 5G, h illustrates the differences in the expression pattern of the two receptors in the root hairs at 3 days post inoculation with *M. loti*. The lower level of expression of NFR1 in the root hairs is clearly visible in spite of the high stability of the triple YFP protein.

Reviewer #3:In this paper, the authors report that an epidermal LysM receptor kinase (LYS1/NFRe) perceives Nod factors and contributes to rhizobial symbiosis, probably by keeping the epidermal cells at an "idling" state. Although the involvement of NFRe in the symbiotic signaling is evident from the results with the corresponding mutants but the detailed mechanism by which NFRe contributes to the symbiosis is still largely obscure.

We thank reviewer #3 for pointing out that our study based on mutant analyses shows clearly the role of NFRe in the symbiosis signalling. We show that Nod factor perception by NFRe is important for regular calcium spiking in the root hairs, for early gene activation after Nod factor treatment and nodule organogenesis. We also show that NFRe and NFR1 kinases have a differential downstream signalling and that the activation loop of the two receptors is the major determinant for this distinct activity. Our findings contribute with important pieces for our understanding of Nod factor signalling and as mentioned by reviewer #2, these will likely change the view of the field on this matter.

Importantly, biochemical characterization of NFRe contains several problems that raise serious questions on the "receptor" function of NFRe.

We present below a detailed response to the comments on the biochemical characterisation of NFRe binding capacities.

The authors analyzed the interaction of the heterologously expressed ectodomains of NFRe and NFR1 using bilayer interferometry, where the biotinylated derivatives of a Nod factor and (GlcNAc)5 were immobilized onto the sensor tip. The results are quite strange. Potato lectin, which is known to bind chitin oligosaccharides but nothing to do with symbiosis, showed a very high affinity (Kd, 30 nM) to the Nod factor, while both NFRe and NFR1 showed thousand times lower (!) affinities to the Nod factor. The affinities of NFRe and NFR1 to the Nod factor (Kd, approximately 30 μM) are also far low compared to the biological activity of Nod factors. In general, the instrument used here seems to give a rather high estimate of the affinity, because the affinity of potato lectin to (GlcNAc)5 obtained here (Kd, 5.2 nM) seems to be much higher than the traditional estimate of the affinity of this lectin to chitin oligosaccharides (The lectins, Academic Press). Thus, it is quite questionable whether such a low affinity of NFRe and NFR1 to the Nod factor observed in this work reflects a biologically significant interaction. I am afraid that such a very low affinity simply reflects some non-specific interaction of these proteins with the hydrophobic region of Nod factor, fatty acid moiety.To avoid such a possibility, the authors should incorporate inhibition type experiments in their binding studies. If the observed binding of NFRe and NFR1 to the Nod factor reflects a biologically significant interaction, it should be inhibited by the biologically active Nod factor but not by inactive Nod factors or chitin oligosaccharides.

We acknowledge and agree with the reviewer that the audience can be misled by the lectin control. The sole purpose of the potato lectin control was to demonstrate correct immobilization and presence of the carbohydrate conjugates on the BLI biosensors, as it is a known binder for N-acetylglucosamine moieties, although its dissociation constant might be affected by avidity. We have exchanged it for a well-studied control, which is the ectodomain of the well-characterized AtCERK1 receptor. The dissociation constant for CO5 to AtCERK1-ECD on BLI was Kd = 59 µM. This value is equal to the previously published dissociation constant Kd = 66 µM for this interaction published by Liu et al., (2012), who used a similar expression and purification protocol and obtained a crystal structure of the AtCERK1-ECD – CO5 complex.

Dissociation constants in the µM concentration range have been observed for multiple LysM receptor kinase ectodomains to various ligands (Liu et al., 2012, Kawaharada et al., 2015, Liu et al., 2016, Bozsoki et al., 2017). Nanomolar affinities have been published for full-length receptors (Iizasa et al., 2010, Broghammer et al., 2012), where avidity between membrane-bound proteins influences the binding affinity. In summary our binding studies are comparable to previous results with LysM ectodomains of plant receptor kinases and the results from nfre mutant studies demonstrate a role in Nod factor signalling.

There are some problems for the characterization of the expressed proteins too. Although the authors described "Proteins of high purity were obtained after four steps of purification, as confirmed by the homogeneous and sharp peaks observed after size exclusion chromatography" (Subsection “NFRe is a Nod factor receptor with an active intracellular kinase”), the results in Figure 1—figure supplement 2B shows the presence of multiple peaks for NFR1-ECD. If the authors used only the peak indicated as NFR1-ECD, they should describe it in the figure legend. It is better to show the elution profiles of final preparations of NFRe/NFR1-ECD by using the same column. The authors should also explain why these preparations showed multiple bands in SDS-PAGE (Figure 1—figure supplement 2C). It is also questionable why the sizes of the smallest bands for NFR1-ECD and NFRe-ECD are very different, while the expected sizes of these proteins are quite similar. I am afraid that some limited proteolysis happened for NFR1-ECD.

We understand the concerns of the reviewer, as the presence of a single band is usually a quality indicator for a protein preparation. Indeed, as correctly noted by the reviewer, the SDS-PAGE shows multiple bands. However, these are not due to limited proteolysis, but instead to different N-glycosylation patterns. LysM receptor kinase ectodomains have multiple disulphide bridges in close proximity and several Nglycosylation sites (see crystal structures in Liu et al., 2012, Liu et al., 2016, Boszoki et al., 2017 as well as Mulder et al. 2006 shows heavy glycosylation for Medicago NFP) and therefore recombinant production requires a eukaryotic expression system capable of providing these modifications to ensure correct folding and good protein quality. This can be seen on the size exclusion profiles, where the ectodomains elute as single peaks, while the void volume peaks contain contaminants. When the recombinant proteins are treated with an endoglycosidase, e.g. PNGase F, the bands collapse to one single, sharp band, clearly demonstrating that the different bands are due to the presence of N-glycans and not to partial degradation. We have followed the reviewer’s suggestions and expanded the figure captions to address N-glycosylations and peaks on the size exclusion profile. We also show an elution profile of NFR1-ECD on the same column as NFRe-ECD.

Characterization of the nfre mutants clearly showed that NFRe contributes to symbiotic signaling and nodule formation. However, the results of the overexpression experiments seem to be difficult to understand. In the nfr1 background, slight increase of the copy number of Nfre gene resulted nothing but the overexpression of the same gene induced a limited symbiotic response induced by M. loti, i.e., activation of Nin promotor in the epidermal cell layer. This Nin activation was not associated with the formation of nodules. Is it safe to connect the results only observed with the overexpression experiments to physiological function? Does the result fit to the hypothesis of "idling" function of NFRe?

We agree with reviewer #3 that caution needs to be taken when interpreting the results from overexpression experiments. In the context of our study, we believe it is important to understand whether NFRe has the capacity to induce Nin induction by Rhizobium when constrains related to the expression level (own or NFR1 promoters) are not present. Furthermore, in order to understand, what is the mechanism behind NFR1 and NFRe differential signalling we have performed direct comparisons between receptors driven by the same promoter. All together the results from biochemical studies of the ectodomain and intracellular kinase, the expression pattern of the Nfre and Nfr1, the results from nfr1 and nfr5 complementation and from the characterisation of the nfre mutant point towards this model where NFRe, most likely together with NFR5, maintains the epidermal layer of the growing root attuned but not committed to symbiosis, state that ensures a rapid switch by NFR1/NFR5 master regulators.

The Venn diagram shown in Figure 3F indicates the presence of significant differences between nfre-1 and nfre-2. Is it OK?

The nfre-1 and nfre-2 are independent alleles that differ in the number and location of LORE1 retroelement insertions, consequently, a difference in the expression pattern is expected. We considered the common differentially expressed genes for comparisons to wild-type, to be NFRe dependent.

Would it be better to confirm the results of nfre mutants by complementation?

We agree that genetic complementation is indeed a powerful tool and based on the nfre phenotype envision that stable genetic lines would be needed. We provide compelling evidence for the role of NFRe in symbiosis based on analyses of three independent alleles which is in accordance with the current accepted standards for genetic analysis.

[Editors’ note: the author responses to the re-review follow.]

Although we acknowledge the technical issue, it is difficult that a method hardly used in the field should explain obvious discrepancies noted by the referees. In this context, it also is somewhat problematic that if the reviewers do not catch it, the majority of readers won't either. Thus, the binding studies should be complemented with additional assays, using a more frequently used reliable.

The main objection towards our manuscript seemed to be based on doubts about the validity of the results from bio-layered interferometry (BLI). The scientific reasoning behind employing BLI approach is that Nod factors, because of the hydrophobic fatty-acid chain and well-known sticky properties, are difficult to work with in solution and need to be immobilized. One major concern from the reviewers was based on the assumption that BLI does not produce binding affinities that are comparable to other methods. To this end, we have now expressed and purified the ectodomain of AtCERK1, which is one of the best studied LysM receptors, and performed BLI measurements to assess binding affinities to chitopentaose and R7A Nod factor, thus providing a previously characterized, published and broadly accepted control for LjNFRe ectodomain binding. We show that AtCERK1 ectodomain has a dissociation constant of K_D_ = 59 µM to chitopentaose and therefore clearly demonstrate that our BLI measurements indeed are robust and comparable to independently published affinity measurements using ITC (Liu et al., 2012; K_D_ = 66 µM)). We would like to stress that the BLI method was already established in 2005 and has now been used in more the 650 published studies including 3 publications in *eLife*.

A revised manuscript has been submitted for your consideration. It contains the aforementioned results on AtCERK1 ectodomain, the activities of Nfr1 and Nfre promoters in the susceptible zone of the root, together with additional revisions according to reviewer’s suggestions.

[Editors’ note: the author responses to the re-review follow.]

1) The highly volatile nature of the spiking frequency makes it very difficult to determine fundamental differences. Thus, more detailed dissection of the patterns is required to support your conclusions. Likewise, strong statements in the text and the abstract should be softened in order to reflect the strength of the phenotype observed in the nfre1-1 mutant.

We agree that analysis of calcium spiking pattern requires careful analysis and interpretation, which we believe it has been done in this manuscript and revealed a quantitative difference between *nfre-1* and wild-type. To this end we have i) investigated the effect of *Nfre* mutation on calcium spiking in stably transformed plants avoiding the interference from a disturbed hormonal balance and growth patterns in hairy roots. ii) investigated a large number of nuclei (50 for wild-type and 46 for the *nfre-1*) to capture the existing diversity in patterns. This is a much larger number than previously used for such detailed microscopy investigations. iii) used an automated software for unbiased analysis of the raw data and interpretation. This type of analysis has been used previously to highlight differences in spiking regularity between MYC- and NOD-induced signals. (Russo et al., 2013)

We have incorporated the reviewer comments into the text that reads now:

Abstract: “Mutants of *Nfre* react to Nod factors with significantly increased calcium spiking interval, reduced transcriptional response and fewer nodules in the presence of rhizobia.” subsection “NFRe perceives Nod factor and has an active intracellular kinase”: “Our findings provide evidence for a complex Nod factor signalling where NFRe activity in the outer root cell layers aids in maintaining a normal calcium spiking interval in the root hairs, integral transcript responses in the susceptible root zone, and initiation of nodule primordia on the expanding root system.”

Subsection “NFRe promotes nodule organogenesis”: “Closer inspection of the spiking frequency revealed that the average inter-spike interval was significantly longer in the *nfre* cells (106s) compared to wild type (86s), indicating that NFRe contributes to a constant interval length of calcium oscillations (Figure 3—figure supplement 2).”

Subsection “The activation segments of NFR1 and NFRe determine the signalling output”: “Together, these results show that NFRe represents an influential component of the epidermal Nod factor signalling in *L. japonicus*, promoting intracellular signalling that leads to optimal calcium spiking, activation of gene transcription and efficient nodule organogenesis on the expanding root system.”

2) The referees believe that (transient) complementation of the nfre1-1 mutant is required to overcome problems associated with quantitatively differing phenotypes observed in the three nfre1-1 mutant alleles.

We have performed the genetic complementation analysis of the *nfre-1* mutant and the results show clearly that the symbiotic phenotype of *nfre-1* is caused by mutation in the *Nfre* gene. These results are now integrated in the revised manuscript Figure 3—figure supplement 1A.

3) The use of inhibition type experiments is mandatory in addition to the differential binding experiments. This is important in order to demonstrate specificity of binding, which in addition to ligand affinity is a major criterion for biologically meaningful receptor-ligand interaction. Use of Nod factor, chitin oligosaccharide or exopolysaccharide as competitors in biolayer interferometry are likely to strengthen your claim about the receptor function of NFRe.You may also wish to show that the biotinylated R7A Nod factor retains the biological activity.

Analyses of receptor-ligand biochemistry in competition assays have been reported so far for successful studies where specificity, or lack of it, could be shown. As correctly pointed by the reviewers, this is an analysis that can provide important information on receptor specificity. We have therefore set out to investigate this in the revision time. We have performed additional expression and purification experiments, but the production of the required amount of protein for competition studies on BLI was not achievable. Our experience based on long term experimentation (J. Cheng- whole PhD study) is that NFR1 and NFRe ectodomains are among the most challenging LysM proteins in terms of expression and purification. We have, nevertheless used the very limited amount of reliable NFRe and NFR1 proteins to perform competition assays based on R7ANod factor/CO5 -coated streptavidin beads in comparison with CERK6, the *Lotus japonicus* chitin receptor (Bozsoki et al., 2017). Unfortunately, the results obtained from two independent experiments are totally inconclusive. We provide the details of our results in the tables below together with representative WB figures as information for editorial and reviewer’s consideration. Our experience is in line with the current literature where no competition binding assays have been reported for any of the ten LysM receptors that have been biochemically and functionally characterised so far in rice, Arabidopsis and Lotus, indicating that such classical competition studies for carbohydrate LysM receptor studies might not resolve an intricate specificity issue as originally thought.

We present here, the most refined binding study for 3 LysM receptors to date, with direct binding data to pure and well characterized Nod factor and chitin ligands. These results show that NFRe and NFR1 ectodomains can bind *M.loti* Nod factor but not CO5 in a biolayer interferometry biochemical assay which is in line with the plant mutant phenotypes showing an impairment in Nod factor but not CO5 signalling. Furthermore, the additional results from the complementation analyses strengthen our conclusion that NFRe is important for Nod factor signalling. The inconclusive results from the competition assays using single receptors point as well towards our proposed model of how complex Nod factor signalling is *in planta*. It is very likely that multiple LysM receptor assemble into functional signalling complexes and that signalling specificity is the result of the nature of the complex, rather than isolated LysM receptors alone. We have incorporated this view more clearly now in the Discussion section of the revised manuscript (Discussion section) to emphasise the complexity of LysM receptor-mediated signalling and revised the following sections as below:

Subsection “NFRe perceives Nod factor and has an active intracellular kinase”. “In short, our study shows that NFRe can bind Nod factor with comparable affinity as seen for the *bona fide* Nod factor receptor NFR1 and both receptor ectodomains distinguish Nod factor from pentameric chitin ligands in a biolayer interferometry assay (Figure 1E).”

Discussion section: “Here, we show that the NFR1-NFR5 signalling cascade operates on the framework provided by the epidermal Nod factor receptor NFRe. NFRe and NFR1 share biochemical and molecular properties i.e similar Nod factor-binding affinity, and chitopentaose differentiating capacity when assessed by biolayer interferometry …”